# ON EPISODES, PROTOTYPICAL NETWORKS, AND FEW-SHOT LEARNING

## ABSTRACT

Episodic learning is a popular practice among researchers and practitioners interested in few-shot learning. It consists of organising training in a series of learning problems, each relying on small "support" and "query" sets to mimic the few-shot circumstances encountered during evaluation. In this paper, we investigate the usefulness of episodic learning in Prototypical Networks, one of the most popular algorithms making use of this practice. Surprisingly, in our experiments we found that, for Prototypical Networks, it is detrimental to use the episodic learning strategy of separating training samples between support and query set, as it is a data-inefficient way to exploit training batches. This "non-episodic" version of Prototypical Networks, which corresponds to the classic Neighbourhood Component Analysis, reliably improves over its episodic counterpart in multiple datasets, achieving an accuracy that is competitive with the state-of-the-art, despite being extremely simple.

## 1 INTRODUCTION

The problem of few-shot learning (FSL) – classifying examples from previously unseen classes given only a handful of training data – has considerably grown in popularity within the machine learning community in the last few years. The reason is likely twofold. First, being able to perform well on FSL problems is important for several applications, from learning new characters (Lake et al., 2015) to drug discovery (Altae-Tran et al., 2017). Second, since the aim of researchers interested in meta-learning is to design systems that can quickly learn novel concepts by generalising from previously encountered learning tasks, FSL benchmarks are often adopted as a practical way to empirically validate meta-learning algorithms.

To the best of our knowledge, there is not a widely recognised definition of meta-learning. In a recent survey, Hospedales et al. (2020) informally describe it as *"the process of improving a learning algorithm over multiple learning episodes"*. Several popular papers in the FSL community (e.g. Vinyals et al. (2016); Ravi & Larochelle (2017); Finn et al. (2017); Snell et al. (2017)) have emphasised the importance of organising training into *episodes*, i.e. learning problems with a limited amount of training and (pseudo-)test examples that resemble the test-time scenario. This popularity has reached such a point that an "episodic" data-loader is often at the core of new FSL algorithms, a practice facilitated by frameworks such as Deleu et al. (2019) and Grefenstette et al. (2019).

Despite the considerable strides made in FSL over the past few years, several recent works (e.g. Chen et al. (2019); Wang et al. (2019); Dhillon et al. (2020); Tian et al. (2020)) showed that simple baselines can outperform established meta-learning methods by using embeddings pre-trained with standard classification losses. These results have cast a doubt in the FSL community on the usefulness of meta-learning and its pervasive episodes. Inspired by these results, we aim at understanding the practical usefulness of episodic learning in arguably the simplest method which makes use of it: Prototypical Networks (Snell et al., 2017). We chose to analyse Prototypical Networks not only for their simplicity, but also because they often appear as important building blocks of newly-proposed methods (e.g. Oreshkin et al. (2018); Cao et al. (2020); Gidaris et al. (2019); Yoon et al. (2019)).

With a set of ablative experiments, we show that for Prototypical Networks episodic learning *a)* is detrimental for performance, *b)* is analogous to randomly discarding examples from a batch and *c)* it introduces a set of unnecessary hyper-parameters that require careful tuning. We also show how, without episodic learning, Prototypical Networks are connected to the classic *Neighbourhood*

*Component Analysis* (NCA) (Goldberger et al., 2005; Salakhutdinov & Hinton, 2007) on deep embeddings. Without bells and whistles, our implementation of the NCA loss achieves an accuracy that is competitive with the state-of-the-art on multiple FSL benchmarks: *mini*ImageNet, CIFAR-FS and *tiered*ImageNet.

## 2 RELATED WORK

Pioneered by the seminal work of Utgoff (1986), Schmidhuber (1987; 1992), Bengio et al. (1992) and Thrun (1996), the general concept of meta-learning is several decades old (for a survey see Vilalta & Drissi (2002); Hospedales et al. (2020)). However, in the last few years it has experienced a surge in popularity, becoming the most used paradigm for learning from very few examples. Several methods addressing the FSL problem by learning on episodes were proposed. MANN (Santoro et al., 2016) uses a Neural Turing Machine (Graves et al., 2014) to save and access the information useful to meta-learn; Bertinetto et al. (2016) propose a deep network in which a "teacher" branch is tasked with predicting the parameters of a "student" branch; Matching Networks (Vinyals et al., 2016) and Prototypical Networks (Snell et al., 2017) are two non-parametric methods in which the contributions of different examples in the support set are weighted by either an LSTM or a softmax over the cosine distances for Matching Networks, and a simple average for Prototypical Networks; Ravi & Larochelle (2017) propose instead to use an LSTM to learn the hyper-parameters of SGD, while MAML (Finn et al., 2017) learns to fine-tune an entire deep network by backpropagating through SGD. Despite these works widely differing in nature, they all stress on the importance of organising training in a series of small learning problems (*episodes*) that are similar to those encountered during inference at test time.

In contrast with this trend, a handful of papers have recently shown that simple approaches that forego episodes and meta-learning can perform well on FSL benchmarks. These methods all have in common that they pre-train a feature extractor with the cross-entropy loss on the "meta-training classes" of the dataset. Then, at test time a classifier is adapted to the support set by weight imprinting (Qi et al., 2018; Dhillon et al., 2020), fine-tuning (Chen et al., 2019), transductive fine-tuning (Dhillon et al., 2020) or logistic regression (Tian et al., 2020). Wang et al. (2019) suggest performing test-time classification by using the label of the closest centroid to the query image.

Different from these papers, we try to shed some light on one of the possible causes behind the poor performance of episodic-based algorithms like Prototypical Networks. An analysis similar to ours in spirit is the one of Raghu et al. (2020). After showing that the efficacy of MAML in FSL is due to the adaptation of the final layer and the "reuse" of the features of previous layers, they propose a variant with the same accuracy and computational advantages. In this paper, we focus on an FSL algorithm just as popular and uncover inefficiencies that allow for a notable conceptual simplification of Prototypical Networks, which surprisingly also brings a significant boost in performance.

## 3 BACKGROUND AND METHOD

This section is divided as follows: Sec. 3.1 introduces episodic learning and the formalism used in FSL, Sec. 3.2 reviews Prototypical Networks (often referred to as PNs from now on), Sec 3.3 describes the classic NCA loss and how exactly it relates to PNs, and Sec. 3.4 explains the three options we explored to perform FSL classification with an NCA-trained feature embedding.

### 3.1 EPISODIC LEARNING

A common strategy to train few-shot learning algorithms is to consider a distribution $\hat{\mathcal{E}}$ over possible subsets of labels that is as close as possible to the one encountered during evaluation $\mathcal{E}$ [1]. Each episodic batch $B_E = \{S, Q\}$ is obtained by first sampling a subset of labels $L$ from $\hat{\mathcal{E}}$, and then sampling images constituting both *support set* $S$ and *query set* $Q$ from the set of images with labels in $L$, where $S = \{(\mathbf{s}_1, y_1), \ldots, (\mathbf{s}_n, y_n)\}$, $Q = \{(\mathbf{q}_1, y_1), \ldots, (\mathbf{q}_m, y_m)\}$, and $S_k$ and $Q_k$ denote the sets of images with label $y = k$ in the support set and query set respectively. In a Maximum Likelihood Estimation framework, training on these episodes can be written (Vinyals et al., 2016) as

---

[1]Note that, in FSL, the sets of classes of training and evaluation are disjoint.

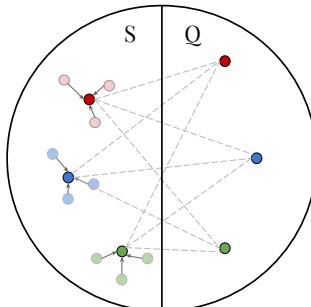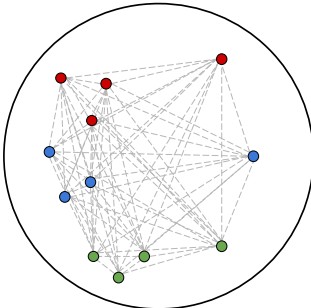

Figure 1: **Batch exploitation** during training for Prototypical Networks (Snell et al., 2017) (left) vs. Neighborhood Component Analysis (Goldberger et al., 2005) (right) on a toy few-shot problem of 3 "ways", 3 "shots" and 1 "query" per class. By dividing batches between support set $S$ and query set $Q$, Prototypical Networks disregards many of the distances between labelled examples that would constitute useful training signal. Details in Sec. 3.3.

the optimisation

$$\arg\max_{\theta} E_{L\sim\hat{\mathcal{E}}} \left[ E_{S\sim L, Q\sim L} \left[ \sum_{(q_i, y_i)\in Q} \log P_\theta\left(y_i | q_i, S\right) \right] \right]. \tag{1}$$

In most implementations this corresponds to training on a series of mini-batches in which each image belongs either to the support or the query set. Support and query sets are constructed such that they both contain all the classes of $L$, and a constant number of images per class. Therefore, episodes are characterised by three values: the number of classes $w = |L|$ (the "ways"), the number of examples per class in the support set $n = |S_k|$ (the "shots"), and the number of examples per class in the query set $m = |Q_k|$. Importantly, during evaluation the triplet $\{w, n, m\}$ defines the problem setup. Usually the triplet remains unchanged across methods, although the Meta-Dataset benchmark of Triantafillou et al. (2019) evaluates on episodes with a variable number of ways and shots. At training time the triplet $\{w, n, m\}$ can be seen as a set of hyper-parameters controlling the batch creation, and that (as we will see in Sec. 4.2) requires careful tuning.

### 3.2 PROTOTYPICAL NETWORKS (PNs)

Prototypical Networks (Snell et al., 2017) are one of the most popular and effective approaches in the few-shot learning literature, and they are at the core of many newly proposed methods (e.g. Oreshkin et al. (2018); Gidaris et al. (2019); Allen et al. (2019); Yoon et al. (2019); Cao et al. (2020)).

During *training*, episodes constituted by support set $S$ and query set $Q$ are sampled as described in Sec. 3.1. Then, a *prototype* for each class $k$ is computed as the mean embedding of the samples from the support set belonging to that class: $\mathbf{c}_k = (1/|S_k|) \cdot \sum_{(\mathbf{s}_i, y_k)\in S_k} f_\theta(\mathbf{s}_i)$, where $f_\theta$ is a deep neural networks with parameters $\theta$ learned via Eq. 1.

Let $C = \{(\mathbf{c}_1, y_1), \ldots, (\mathbf{c}_k, y_k)\}$ be the set of prototypes and corresponding labels. The loss can be written as follows:

$$\mathcal{L}_{\text{proto}}(S, Q) = -\frac{1}{|Q|} \sum_{(\mathbf{q}_i, y_i)\in Q} \log\left( \frac{\exp -\|f_\theta(\mathbf{q}_i) - \mathbf{c}_{y_i}\|^2}{\sum_{k'} \exp -\|f_\theta(\mathbf{q}_i) - \mathbf{c}_{k'}\|^2} \right). \tag{2}$$

Here, $k'$ is an index that goes over all classes. This loss is minimised over a number of training episodes. After training, given a query image $\mathbf{x}_i$ from a new test episode, classification is conducted by simply consulting the nearest-neighboring prototype computed from the support set of the episode, i.e. $y(\mathbf{x}_i) = \arg\min_{j\in\{1,\ldots,k\}} \|f_\theta(\mathbf{x}_i) - \mathbf{c}_j\|$.

### 3.3 NEIGHBOURHOOD COMPONENT ANALYSIS (NCA)

Eq. 2 computes the likelihood that a query image belongs to the class a certain prototype is representative of by computing the softmax over the distances to all prototypes. This formulation is similar to

|  | positives | negatives |
|---|---|---|
| **PNs** | $wmn$ | $w(w-1)mn$ |
| **NCA** | $\binom{m+n}{2}w$ | $\binom{w}{2}(m+n)^2$ |

Table 1: Number of gradients from positive and negative distance pairs contributing to the loss within a batch for NCA and PNs. Respectively, $w$, $n$ and $m$ represent the number of ways, shots and queries. In Appendix A.8 we show that the extra number of pairs NCA can exploit grows as $O(w^2(m^2 + n^2))$.

the *Neighbourhood Component Analysis* approach by Goldberger et al. (2005) (and expanded to the non-linear case by Salakhutdinov & Hinton (2007)), except for a few important differences which we will now discuss.

Let $i \in [1, b]$ be the indices of the images within a batch $B$. The NCA loss can be written as:

$$\mathcal{L}_{\text{NCA}}(X) = -\frac{1}{b} \sum_{i \in 1,\ldots,b} \log \left( \frac{\sum_{\substack{j \in 1,\ldots,b \\ j \neq i \\ y_i = y_j}} \exp - \left\| \mathbf{z}_i - \mathbf{z}_j \right\|^2}{\sum_{\substack{k \in 1,\ldots,b \\ k \neq i}} \exp - \left\| \mathbf{z}_i - \mathbf{z}_k \right\|^2} \right), \tag{3}$$

where $\mathbf{z}_i = f_\theta(\mathbf{x}_i)$ is an image embedding and $y_i$ its corresponding label. By minimising this loss, distances between embeddings from the same class will be minimised, while distances between embeddings from different classes will be maximised. This bears similarities to the (supervised) contrastive loss (Khosla et al., 2020; Chen et al., 2020a), which we discuss in Appendix A.3. For ease of discussion, we refer to the distances between pairs of embeddings from the same class as *positives*, and to the distances between pairs of embeddings from different classes as *negatives*.

Importantly, the concepts of support set and query set of Sec. 3.1 and 3.2 here do not apply. More simply, the images (and respective labels) constituting the batch $B = \{(\mathbf{x}_1, y_1), \ldots, (\mathbf{x}_b, y_b)\}$ are sampled i.i.d. from the training dataset, as normally happens in standard supervised learning. Considering that in PNs there is no parameter adaptation happening at the level of each episodic batch $B_E$, $S$ and $Q$ *do not* have a functional role in the algorithm, and their defining values $\{w, m, n\}$ can be interpreted as hyper-parameters controlling the data-loader during training. More specifically, PNs differ from NCA in three key aspects:

1. PNs rely on the creation of prototypes, while NCA does not.
2. Due to the nature of episodic learning, PNs only consider pairwise distances between the query and the support set; the NCA instead uses *all* the distances within a batch and treats each example in the same way.
3. Because of how $L$, $S$ and $Q$ are sampled in episodic learning, some images will be inevitably sampled more frequently than others, and some will likely never be seen during training (this corresponds to sampling "with replacement"). NCA instead visits every image of the dataset once and only once within each epoch (sampling "without replacement").

**Notes on data-efficiency in batch exploitation.** To expand on point 2 above, Fig. 1 illustrates the difference in batch exploitation. For PNs (*left*) and a training episode of $w$=3 ways, $n$=3 shots and $m$=1 queries, the total number distances contributing to the loss consists of $wmn = 9$ positives and $w(w-1)mn = 18$ negatives [2]. If we consider the *same* training batch in a nonepisodic way, when computing the NCA loss (Fig. 1 *right*) we have $\binom{m+n}{2}w = 18$ positives and $\binom{w}{2}(m+n)^2 = 48$ negatives (we summarise these equations in Table 1). This difference in batch exploitation is significant; in Appendix A.8 we show that it grows exponentially as $O(w^2(m^2+n^2))$.

To investigate the effect of the three key differences between PNs and NCA illustrated in this section, in Sec. 4 we conduct a wide range of experiments.

---

[2]The number of distances computed in the batch is actually even smaller for PNs: $wm$=3 positives and $w(w-1)m$=6 negatives. However, since gradients are propagated through prototypes, the distances to the individual support points can be considered as contributing to the loss. This is what we do throughout the paper.

### 3.4 FEW-SHOT CLASSIFICATION DURING EVALUATION

Once $f_\theta$ has been trained, there are many possible ways to perform few-shot classification during evaluation. In this paper we consider three simple approaches that are particularly aligned for embeddings learned via metric-based losses such as Eq. 2 or Eq. 3.

$k$**-NN.** To classify an image $\mathbf{q}_i \in Q$, we first compute the Euclidean distance to each support point $\mathbf{s}_j \in S$: $d_{ij} = \|f_\theta(\mathbf{q}_i) - f_\theta(\mathbf{s}_j))\|^2$. Then, we simply assign $y(\mathbf{q}_i)$ to be majority label of the $k$ nearest neighbours. A downside here is that $k$ is a hyper-parameter that has to be chosen, although a reasonable choice in the FSL setup is to set it equal to the number of "shots" $n$.

$1$**-NN with Class Centroids.** Similar to $k$-NN, we can perform classification by inheriting the label of the closest class centroid, i.e. $y(\mathbf{q}_i) = \arg\min_{j \in \{1,...,k\}} \|f_\theta(\mathbf{x}_i) - \mathbf{c}_j\|$. This is the approach used at test-time by Snell et al. (2017) and Wang et al. (2019).

**Soft Assignments.** This is what the original NCA paper (Goldberger et al., 2005) used for evaluation. To classify an image $\mathbf{q}_i \in Q$, we compute the values $p_{ij} = \exp(-\|f_\theta(\mathbf{q}_i) - f_\theta(\mathbf{s}_j))\|^2)/\sum_{\mathbf{s}_k \in S} \exp(-\|f_\theta(\mathbf{q}_i) - f_\theta(\mathbf{s}_k)\|^2)$ for all $\mathbf{s}_j \in S$, which is the probability that image $i$ is sampled from image $j$. We then compute the likelihood for each class $k$: $\sum_{s_j \in S_k} p_{ij}$, and choose the class with the highest likelihood $y(\mathbf{q}_i) = \arg\max_k \sum_{s_j \in S_k} p_{ij}$. This approach is the only one closely aligned with the training procedure, and has a direct probabilistic interpretation.

We also experimented with using the NCA loss on the support set to perform adaptation at test time, which we discuss in Appendix A.1.

## 4 EXPERIMENTS

In the following, Sec. 4.1 describes our experimental setup. Sec. 4.2 shows the effect of the hyper-parameters controlling the creation of episodes in PNs. In Sec. 4.3 we perform a set of ablation studies to better illustrate the relationship between PNs and the NCA. Finally, in Sec. 4.4 we compare our version of the NCA to several recent methods on three FSL benchmarks.

### 4.1 EXPERIMENTAL SETUP

We conduct our experiments on *mini*ImageNet (Vinyals et al., 2016), CIFAR-FS (Bertinetto et al., 2019) and *tiered*ImageNet (Ren et al., 2018), using the ResNet12 variant first adopted by Lee et al. (2019) as embedding function $f_\theta$. A detailed description of benchmarks, architecture and implementation details is deferred to Appendix A.4, while below we discuss the most important choices of the experimental setup.

Like Wang et al. (2019), for all our experiments (including those with Prototypical Networks) we centre and normalise the feature embeddings before performing classification, as it is considerably beneficial for performance. After training, we compute the mean feature vectors of all the images in the training set: $\bar{\mathbf{x}} = \frac{1}{|\mathcal{D}^{\text{train}}|} \sum_{\mathbf{x} \in \mathcal{D}^{train}} \mathbf{x}$. Then, all feature vectors in the test set are updated as $\mathbf{x}_i \leftarrow \mathbf{x}_i - \bar{\mathbf{x}}$, and normalised by $\mathbf{x}_i \leftarrow \frac{\mathbf{x}_i}{\|\mathbf{x}_i\|}$.

We compared the inference methods discussed in Sec. 3.4 on *mini*ImageNet and CIFAR-FS. Results can be found in Table 2. We chose to use 1-NN with class centroids in all our experiments, as it performs significantly better than $k$-NN or Soft Assignment. This might sound surprising, as the Soft Assignment approach closely reflects the training protocol and outputs class probabilities. We speculate its inferior performance could be caused by poor model calibration (Guo et al., 2017): since the classes between training and evaluation are disjoint, the model is unlikely to produce calibrated probabilities. As such, within the softmax, outliers behaving as false positives can happen to highly influence the final decision, and those behaving as false negatives can end up being almost completely ignored (their contribution is squashed toward zero). With the nearest centroid classification approach outliers are still clearly an issue, but their effect can be less dramatic.

As standard, performance is assessed on episodes of 5-way, 15-query and either 1- or 5-shot. Each model is evaluated on 10,000 episodes sampled from the test set (or the validation set, in some

|        |  *mini*ImageNet  |              |
|--------|------------------|--------------|
| method | 5-shot val       | 5-shot test  |
| Soft Assignment | $79.11 \pm 0.27$ | $77.16 \pm 0.10$ |
| $k$-NN | $75.82 \pm 0.21$ | $73.52 \pm 0.12$ |
| 1-NN centroid | $\mathbf{80.61 \pm 0.20}$ | $\mathbf{78.30 \pm 0.14}$ |
|        | CIFAR-FS | |
| Soft Assignment | $76.12 \pm 0.32$ | $83.31 \pm 0.37$ |
| $k$-NN | $73.46 \pm 0.39$ | $80.94 \pm 0.38$ |
| 1-NN centroid | $\mathbf{77.80 \pm 0.35}$ | $\mathbf{85.13 \pm 0.32}$ |

Table 2: Comparison between the different evaluation methods discussed in Sec. 3.4.

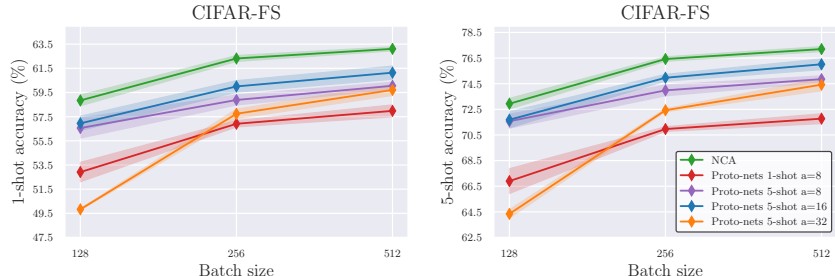

Figure 2: 1-shot (left) and 5-shot accuracies (right) on the val. set of CIFAR-FS. Models trained using NCA, or Proto-nets with different configurations: 1-shot with $a$=8 and 5-shot with $a$=8, 16 or 32. Values correspond to the mean accuracy of five models trained with different random seeds. Please see Sec. 4.2 for details.

experiments). To further reduce the variance, we trained each model five times with five different random seeds, for a total of 50,000 episodes per configuration, from which error bars are computed.

## 4.2 CONSIDERATIONS ON EPISODES AND DATA EFFICIENCY

Despite Prototypical Networks being one of the simplest FSL methods, the creation of episodes requires the use of several hyper-parameters ($\{w, m, n\}$, Sec. 3.1) which can significantly affect performance. Snell et al. (2017) state that the number of shots $n$ between training and testing should match and that one should use a higher number of ways $w$ during training time. In their experiments, they train 1-shot models with $w = 30$, $n = 1$, $m = 15$ and 5-shot models with $w = 20$, $n = 5$, $m = 15$. This makes the corresponding batch sizes of these episodes 480 and 400, respectively. Since, as seen in Sec. 3.3, the number of positives and negatives grows rapidly for both PNs and NCA (although at a different rate), this makes a fair comparison between models trained on different types of episodes difficult.

We investigate the effect of changing these hyper-parameters in a systematic manner. To compare configurations fairly across episode/batch sizes, we define each configuration by its number of shots $n$, the batch size $b$ and the total number of images per class $a$ (which accounts for the sum between support and query set, $a = n + m$). For example, if we train a 5-shot model with $a = 8$ and $b = 256$, it means that its corresponding training episodes will have $n = 5$, $q = 8 - 5 = 3$, and $w = 256/8 = 32$. Using this notation, we train configurations of PNs covering several combinations of these hyper-parameters so that the resulting batch size corresponds to an episode is 128, 256 or 512. Then, we train three configurations of NCA, where the only hyper-parameter is the batch size.

Results for CIFAR-FS can be found in Fig. 2, where we report results for NCA and PNs with $a = 8$, 16 or 32. Results for *mini*ImageNet observe the same trend and are deferred to Appendix A.6. Note that the results of 1-shot with $a = 16$ and $a = 32$ are not reported, as they fare significantly worse. Several things can be noticed. First, NCA performs better than *all* PNs configurations, no matter the batch size. Second, PNs is very sensitive to different hyper-parameter configurations. For instance, with batches of size 128, PNs trained with episodes of 5-shot and $a = 32$ performs significantly worse than a PNs trained with episodes of 5-shot and $a = 16$. Finally, we can also notice that, contrary to what has been previously reported (Snell et al., 2017; Cao et al., 2020), the 5-shot model with the best configuration is always strictly better than any 1-shot configuration. We speculate that this is probably due to the fact that we compare configurations keeping the batch size constant.

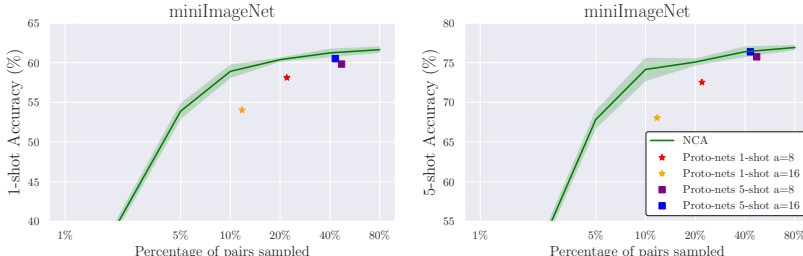

Figure 3: 1-shot and 5-shot accuracies on the *mini*ImageNet test set for models trained with a batch size of 256 while only sampling a % of the total number of available pairs. Reported values correspond to the mean accuracy of five models trained with different random seeds. Individual points contain models trained using PNs which are plotted on the x-axis based on the relative percentage of distance pairs that are used in their computation compared to NCA on the same batch size. Please see Sec. 4.2 for details.

**Episodic batches vs random sub-sampling of standard batches.** Despite the NCA outperforming all PNs configurations (Fig. 2), one might posit that by using more distances within a batch, NCA is more computationally costly and thus the episodic strategy of PNs can sometimes be advantageous (e.g. real-time applications with large batches). To investigate this, we perform an experiment where we train NCA models by randomly sampling a fraction of the total number of distances used in the loss. Then, for comparison, we include different PNs models after having computed to which percentage of discarded pairs (in a normal batch) their episodic batch corresponds to.

Results can be found in Fig. 3. As expected, we can see how sub-sampling a fraction of the total available number of pairs within a batch negatively affects performance. To answer to possible concerns on computational efficiency, we can notice that the PNs points lie very close to the under-sampling version of the NCA, which signals that the episodic strategy of PNs is roughly equivalent to train with the NCA and only exploiting a fraction of the distances available in a batch.

We see as we move right on the curve in Fig. 3, that the PN performance increases as well. In Appendix A.9 we look more carefully whether the difference in performance between the different episodic batch setups of Fig. 2 can be explained by the differences in the number of distance pairs used in the batch configurations. We indeed find that generally speaking the higher the number of pairs the better, however, one should also consider the positive/negative balance and the number of classes present within a batch.

## 4.3 ABLATION EXPERIMENTS

To better analyse why NCA performs better than PNs, in this section we consider the three key differences discussed earlier by performing a series of ablations on models trained on batches of size 256. Results are summarised in Fig. 4. We refer the reader to Appendix A.1 to obtain detailed steps describing how these ablations affect Eq. 2 and Eq. 3, while Appendix A.7 contains the same analysis also for batches of size 128 and 512.

First, we compare two variants of NCA: one in which the sampling of the training batches happens sequentially and without replacement, as it is standard in supervised learning, and one where the batches are sampled with replacement. Interestingly, this modification (row 1 and 2 of Fig. 4) has a negligible effect across the two datasets and four splits considered, meaning that the replacement sampling introduced by episodic learning will not interfere with the other ablations.

We then perform a series of ablations on episodic batches, i.e. sampled with the method described in Sec. 3.1. For each ablation, we perform experiments on PNs trained with 1-shot and 5-shot, both with $a = 8$. This means that both the 1-shot and 5-shot models have 32 classes and 8 images per class, allowing a fair comparison. The batch size is 256 for the NCA too. We first train standard PNs models. Next, we train a PNs model where "prototypes" are not computed (point 1 of Sec. 3.3), meaning that distances are considered between individual points, but a separation between query and support set remains. Then, we perform an ablation where we ignore the separation between support and query set (point 2 of Sec. 3.3), and compute the NCA on the union of the support and query set, while still computing prototypes for the points that would belong to the support set. Last, we

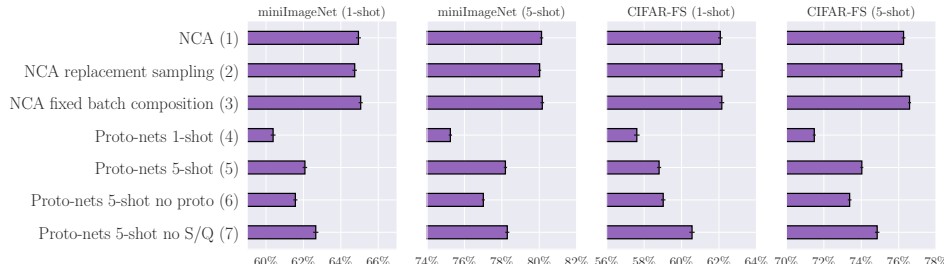

Figure 4: Ablation experiments on NCA and Prototypical Networks, both on batches or episodes of size 256 on the validation set of *mini*ImageNet and CIFAR-FS. Please refer to Sec. 4.3 for details.

perform an ablation where we consider all the previous points together: we sample with replacement, we ignore the separation between support and query set and we do not compute prototypes. This amounts to the NCA loss, except that it is computed on batches with a fixed number of classes and a fixed number of images per class. Notice that in Fig. 4 there is only one row dedicated to 1-shot models. This is because we cannot generate prototypes from 1-shot models, so we cannot have a "no proto" ablation. Furthermore, for 1-shot models the "no S/Q" ablation is equivalent to the NCA with a fixed batch composition. From Fig. 4, we can see that disabling prototypes (row 6) negatively affects the performance of 5-shot (row 5), albeit slightly. On the other hand, enabling the computation between all pairs increases the performance (last row), Importantly, enabling all the ablations (row 3) completely recovers the performance lost by PNs.

The fact that each single ablation does not have much influence on the performance, but their combination does, could be explained by the number of distance pairs exploited by the individual ablations. Using the formulas described at the end of Section 3.3, we compute the number of positives and negatives used for each ablation. For row 6 there are 480 positives, and 14,880 negatives. For row 7 there are 576 positives and 20,170 negatives. In both cases, the number is significantly lower than the corresponding NCA, which gets 896 positives and 31,744 negatives, and this could explain the jump in performance from row 6/7 to 3. Moreover, from row 5/6 to 7 we see a slight increase in performance, which can also be explained by the (slightly) larger number of distance pairs.

These experiments nonetheless highlight that the separation of roles between the images belonging to support and query set, which is typical of episodic learning (Vinyals et al., 2016), is detrimental for the performance of Prototypical Networks. Instead, using the NCA loss on standard mini-batches allows full exploitation of the training data and significantly improves performance. Moreover, the NCA has the advantage of simplifying the overall training procedure, as the hyper-parameters for the creation of episodes $\{w, n, m\}$ no longer need to be considered.

Additionally, in Appendix A.7 we repeat the same analysis done here to the simplest variant of Matching Networks (Vinyals et al., 2016) and obtain the same conclusion; also in this case, the separation of roles between support and query samples severely affects performance.

## 4.4 COMPARISON WITH THE STATE-OF-THE-ART

We now benchmark our models on three FSL datasets, with the purpose of contextualising their performance against the modern literature.

When considering which methods to compare against, we chose those *a)* which have been recently published, *b)* that are using a ResNet12 (which we found the most commonly used) and *c)* with a setup that is not significantly more complicated than ours. For example, we only report the results of the main approach proposed by Tian et al. (2020) and not their *sequential self-distillation* (Furlanello et al., 2018) variant, which requires re-training multiple times and can be applied to most methods.

Results can be found in Table 3 for *mini*ImageNet and CIFAR-FS and Table 4 for *tiered*ImageNet. In Table 3 we report Prototypical Networks results for both the episodic setup from Snell et al. (2017) and the best one (batch size 512, 5-shot, $a{=}16$) found from the experiment of Fig. 2, which brings a considerable improvement over the original. We did not optimised for a new setup for Prototypical Networks on *tiered*ImageNet, as the larger dataset and the higher number of classes would have made the hyper-parameter search too demanding. Notice how our vanilla NCA is competi-

| | *mini*ImageNet | | CIFAR-FS | |
|---|---|---|---|---|
| | **1-shot** | **5-shot** | **1-shot** | **5-shot** |
| **Episodic methods** | | | | |
| adaResNet (Munkhdalai et al., 2018) | $56.88 \pm 0.62$ | $71.94 \pm 0.57$ | - | - |
| TADAM(Oreshkin et al., 2018) | $58.50 \pm 0.30$ | $76.70 \pm 0.30$ | - | - |
| Shot-Free (Ravichandran et al., 2019) | $60.71\pm$ n/a | $77.64\pm$ n/a | $69.2\pm$ n/a | $84.7\pm$ n/a |
| TEAM (Qiao et al., 2019) | $60.07\pm$ n/a | $75.90\pm$ n/a | - | - |
| MTL (Sun et al., 2019) | $61.20 \pm 1.80$ | $75.50 \pm 0.80$ | - | - |
| TapNet (Yoon et al., 2019) | $61.65 \pm 0.15$ | $76.36 \pm 0.10$ | - | - |
| MetaOptNet-SVM(Lee et al., 2019) | $62.64 \pm 0.61$ | $78.63 \pm 0.46$ | $\mathbf{72.0 \pm 0.7}$ | $84.2 \pm 0.5$ |
| Variatonal FSL (Zhang et al., 2019) | $61.23 \pm 0.26$ | $77.69 \pm 0.17$ | - | - |
| **Simple baselines** | | | | |
| Transductive finetuning (Dhillon et al., 2020) | $62.35 \pm 0.66$ | $74.53 \pm 0.54$ | $70.76 \pm 0.74$ | $81.56 \pm 0.53$ |
| RFIC-simple (Tian et al., 2020) | $62.02 \pm 0.63$ | $\mathbf{79.64 \pm 0.44}$ | $71.5 \pm 0.8$ | $86.0 \pm 0.5$ |
| Meta-Baseline (Chen et al., 2020b) | $\mathbf{63.17 \pm 0.23}$ | $79.26 \pm 0.17$ | - | - |
| *Our implementations:* | | | | |
| Proto-nets (Snell et al. (2017) setup) | $59.93 \pm 0.23$ | $75.89 \pm 0.16$ | $70.20 \pm 0.22$ | $83.96 \pm 0.16$ |
| Proto-nets (our setup) | $61.32 \pm 0.23$ | $77.77 \pm 0.15$ | $70.41 \pm 0.31$ | $84.46 \pm 0.29$ |
| SimpleShot (Wang et al., 2019) | $62.16 \pm 0.23$ | $78.33 \pm 0.17$ | $70.01 \pm 0.21$ | $84.50 \pm 0.11$ |
| **NCA (ours)** | $62.52 \pm 0.24$ | $78.3 \pm 0.14$ | $\mathbf{72.48 \pm 0.40}$ | $\mathbf{85.13 \pm 0.29}$ |

Table 3: Comparison of methods that use ResNet12 on *mini*ImageNet and CIFAR-FS (test set).

| | *tiered*ImageNet | |
|---|---|---|
| **Method** | **1-shot** | **5-shot** |
| Shot-Free (Ravichandran et al., 2019) | $63.52\pm$ n/a | $82.59\pm$ n/a |
| RFIC-simple (Tian et al., 2020) | $\mathbf{69.74 \pm 0.72}$ | $\mathbf{84.41 \pm 0.55}$ |
| Meta-Baseline (Chen et al., 2020b) | $68.62 \pm 0.27$ | $83.29 \pm 0.18$ |
| MetaOptNet-SVM(Lee et al., 2019) | $65.99 \pm 0.72$ | $81.56 \pm 0.53$ |
| *Our implementations:* | | |
| Proto-nets (Snell et al. (2017) setup) | $65.45 \pm 0.23$ | $81.14 \pm 0.17$ |
| SimpleShot (Wang et al., 2019) | $66.17 \pm 0.15$ | $80.64 \pm 0.20$ |
| **NCA (ours)** | $68.36 \pm 0.11$ | $83.20 \pm 0.18$ |

Table 4: Comparison of methods that use ResNet12 on *tiered*ImageNet (test set).

tive or superior to recent methods, despite being extremely simple. It fairs surprisingly well against methods that use meta-learning (and episodic learning), and also against the high-performing simple baselines based on pre-training with the cross-entropy loss.

## 5 CONCLUSION

Towards the aim of understanding the reasons behind the poor competitiveness of meta-learning methods with respect to simple baselines, in this paper we start by investigating the role of episodes in the popular Prototypical Networks. We found that their performance is highly sensitive to the set of hyper-parameters used to sample the episodes. By replacing the Prototypical Networks' loss with the classic Neighbourhood Component Analysis, we are able to ignore these hyper-parameters while significantly improving the few-shot classification accuracy. With a series of experiments, we found out that the performance discrepancy mostly arises from the separation between support and query set within each episode, and that Prototypical Networks' episodic strategy is almost empirically equivalent to randomly discarding a large fraction of distances within a standard mini-batch. Finally, we show that our variant of the NCA achieves an accuracy on multiple popular FSL benchmarks that is comparable or superior with state-of-the-art methods of similar complexity, making it a simple and appealing baseline for future work.

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

# A APPENDIX

## A.1 PERFORMANCE IMPROVEMENTS

**Adapting to the support set.** Prototypical Networks does not perform any kind of parameter adaptation at test time. On the one hand this is convenient, as it allows fast inference; on the other hand, useful information from the support set $S$ might remain unexploited.

In the 5-shot case it is possible to minimise the NCA loss since it can directly be computed on the support set: $\mathcal{L}_{\text{NCA}}(S)$. We tried training a positive semi-definite matrix $A$ on the outputs of the trained neural network, which corresponds to learning a Mahalanobis distance metric as in Goldberger et al. (2005). However, we found that there was no meaningful increase in performance. Differently, we did find that fine-tuning the whole neural network $f_\theta$ by $\arg\min_\theta \mathcal{L}_{\text{NCA}}(S)$ was beneficial (see Table 5). However, given the computational cost, we opted for non performing adaptation to the support sets in our experiments of Sec. 4.

**Features concatenation.** For NCA, we also found that concatenating the output of intermediate layers modestly improves performance at (almost) no additional cost. We used the output of the average pool layers from all ResNet blocks except the first and we refer to this variant as NCA multi-layer. However, since this is an orthogonal variation that can be applied to several methods, we do not consider it for our experiments of Sec. 4.

Results on *mini*ImageNet an CIFAR-FS are shown in Table 5.

## A.2 DETAILS ABOUT THE ABLATION STUDIES OF SECTION 4.3

Referring to the three key differences between the Prototypical Networks and the NCA losses listed in Sec. 3.3, in this section we detail how to obtain the ablations we used to perform the experiments of Sec. 4.3.

We can "disable" the creation of prototypes (point 1), which will change the prototypical loss of Eq. 2 to

$$\mathcal{L}(S,Q) = -\frac{1}{|Q|+|S|} \sum_{(\mathbf{q}_i,y)\in Q} \log \left( \frac{\sum\limits_{(\mathbf{s}_j,y')\in S_y} \exp-\|\mathbf{q}_i - \mathbf{s}_j\|^2}{\sum\limits_{(\mathbf{s}_k,y'')\in S} \exp-\|\mathbf{q}_i - \mathbf{s}_k\|^2} \right). \tag{4}$$

This is similar to $\mathcal{L}_{\text{NCA}}$ (Eq. 3), where the positives are represented by the distances from $Q$ to $S_k$, and the negatives by the distances from $Q$ to $S \setminus S_k$. The only difference now is the separation of the batch into a query and support set.

Independently, we can "disable" point 2, which gives us

$$\mathcal{L}(S,Q) = -\frac{1}{|Q|+|S|} \sum_{(\mathbf{z}_i,y_i)\in Q\cup C} \log \left( \frac{\sum\limits_{\substack{(\mathbf{z}_j,y_j)\in Q\cup C \\ y_j=y_i \\ i\neq j}} \exp-\|\mathbf{z}_i - \mathbf{z}_j\|^2}{\sum\limits_{\substack{(\mathbf{z}_k,y_k)\in Q\cup C \\ k\neq i}} \exp-\|\mathbf{z}_i - \mathbf{z}_k\|^2} \right), \tag{5}$$

which essentially combines the prototypes with the query set, and computes the NCA loss on that total set of embeddings.

Finally, we can "disable" both point 1 and 2, which gives us

$$\mathcal{L}(S,Q) = -\frac{1}{|Q|+|S|} \sum_{(\mathbf{z}_i,y_i)\in Q\cup S} \log \left( \frac{\sum\limits_{\substack{(\mathbf{z}_j,y_j)\in Q\cup S \\ y_j=y_i \\ i\neq j}} \exp-\|\mathbf{z}_i - \mathbf{z}_j\|^2}{\sum\limits_{\substack{(\mathbf{z}_k,y_k)\in Q\cup S \\ k\neq i}} \exp-\|\mathbf{z}_i - \mathbf{z}_k\|^2} \right). \tag{6}$$

This almost exactly corresponds to the NCA loss, where the only difference is the construction of batches with a fixed number of classes and a fixed number of images per class.

|  | *mini*ImageNet | | CIFAR-FS | |
| --- | --- | --- | --- | --- |
| method | 1-shot | 5-shot | 1-shot | 5-shot |
| NCA | $62.52 \pm 0.24$ | $78.3 \pm 0.14$ | $72.48 \pm 0.40$ | $85.13 \pm 0.29$ |
| NCA multi-layer | $63.21 \pm 0.08$ | $79.27 \pm 0.08$ | $72.44 \pm 0.36$ | $85.42 \pm 0.29$ |
| NCA (ours) multi-layer + ss | - | $\mathbf{79.79 \pm 0.08}$ | - | $\mathbf{85.66 \pm 0.32}$ |

Table 5: Comparison between vanilla NCA, NCA using multiple evaluation layers and NCA performing optimisation on the support set (ss). The NCA can only be optimised in the 5-shot case, since there are not enough positives distances in the 1-shot case. Support set is optimised for 5 epochs using Adam with learning rate 0.0001 and weight decay 0.0005. For details, see Sec. A.1

## A.3 DIFFERENCES BETWEEN THE NCA AND CONTRASTIVE LOSSES

Eq. 3 is similar to the contrastive loss functions (Khosla et al., 2020; Chen et al., 2020a) that are used in self-supervised learning and representation learning. The main differences are that 1.) In contrastive losses, the denominator only contains negative pairs and 2.) the inner sum in the numerator is moved outside of the logarithm in the supervised contrastive loss function from Khosla et al. (2020). We opted to work with the NCA loss because we found it performs better than the supervised constrastive loss in a few-shot learning setting. Using the supervised contrastive loss we only managed to obtain 51.05% 1-shot and 63.36% 5-shot performance on the *mini*Imagenet test set.

## A.4 IMPLEMENTATION DETAILS

**Benchmarks.** In our experiments, we use three popular FSL benchmarks. ***mini*ImageNet** (Vinyals et al., 2016) is a subset of ImageNet generated by randomly sampling 100 classes, each with 600 randomly sampled images. We adopt the commonly used splits of Ravi & Larochelle (2017) who use 64 classes for meta-training, 16 for meta-validation and 20 for meta-testing. **CIFAR-FS** was proposed by Bertinetto et al. (2019) as an anagolous version of *mini*Imagenet for CIFAR-100. It uses the same sized splits and same number of images per split as *mini*ImageNet. ***tiered*ImageNet** (Ren et al., 2018) is also constructed from ImageNet, but contains 608 classes, with 351 training classes, 97 validation classes and 160 test classes. The class split have been generated using WordNet (Miller, 1995) to ensure that the training classes are semantically "distant" to the validation and test classes. For all datasets, we use images of size $84 \times 84$.

**Architecture.** In all our experiments, $f_\theta$ is represented by a ResNet12 with widths $[64, 160, 320, 640]$. We chose this architecture, initially introduced by Lee et al. (2019), as it is the one which is most frequently adopted by recent FSL methods. Unlike most methods, we do not use a DropBlock regulariser (Ghiasi et al., 2018), as we did not notice it to meaningfully contribute to performance.

**Optimisation.** To train all the models used for our experiments, unless differently specified, we used a SGD optimiser with Nesterov momentum, weight decay of 0.0005 and initial learning rate of 0.1. For *mini*ImageNet and CIFAR-FS we decrease the learning rate by a factor of 10 after 70% of epochs have been trained, and train for a total of 120 epochs. As data augmentations, we use random horizontal flipping and centre cropping.

Only for the experiments of Sec. 4.4, we slightly change our training setup. On CIFAR-FS, we increase the number of training epochs from 120 to 240, which improved accuracy of about 0.5%. For *tiered*ImageNet, we train for 120 epochs and decrease the learning rate by a factor of 10 after 50% and 75% of the training progress. For *tiered*ImageNet only we increased the batch size to 1024, as we found it being beneficial. For the other datasets it did not improve performance. These changes affect all our methods and baselines: NCA, Prototypical Networks (with both old and new batch setup), and SimpleShot (Wang et al., 2019).

**Projection network.** Similarly to (Khosla et al., 2020; Chen et al., 2020a), we also experimented (for both PNs and NCA) with a *projection network* (but *only* for the comparison of Sec. 4.4). The projection network is a single linear layer $A \in \mathbb{R}^{M \times P}$ that is placed on top of $f_\theta$ at training time,

|  | *mini*ImageNet | | CIFAR-FS | |
| --- | --- | --- | --- | --- |
| method | 1-shot | 5-shot | 1-shot | 5-shot |
| PNs (SimpleShot) | $57.99 \pm 0.21$ | $74.33 \pm 0.16$ | $53.76 \pm 0.22$ | $68.54 \pm 0.19$ |
| PNs (ours) | $62.79 \pm 0.12$ | $78.82 \pm 0.09$ | $59.60 \pm 0.13$ | $74.64 \pm 0.11$ |
| NCA (SimpleShot) | $61.21 \pm 0.22$ | $76.39 \pm 0.16$ | $59.41 \pm 0.24$ | $73.29 \pm 0.19$ |
| NCA (ours) | $64.94 \pm 0.13$ | $80.12 \pm 0.09$ | $62.07 \pm 0.14$ | $76.26 \pm 0.10$ |

Table 6: Comparison of results on validation set of *mini*ImageNet and CIFAR-FS using the hyperparameters used in SimpleShot(Wang et al., 2019) and the hyperparameters used in this paper (**ours**). Results are on batch size 256 (as used in Wang et al. (2019)) with the PNs episodic batch being $a=16$, as it is the best performing episodic setup we found.

where $M$ is the output dimension of the neural network $f_\theta$ and $P$ is the output dimension of $A$, which can be considered as a hyper-parameter. The output of $A$ is only used during training. At test time, we do not use the output of $A$ and directly use the output of $f_\theta$. For CIFAR-FS and *tiered*ImageNet, we found this did not help performance. For *mini*ImageNet however we found that this improved performance, and we set $P = 128$ (which worked best for both PNs and NCA). Note that this is not an unfair advantage over other methods. Compared to SimpleShot (Wang et al., 2019) and other simple baselines, we actually use fewer parameters without the projection network (effectively making our ResNet12 a ResNet11) since they use an extra fully connected layer to minimise cross entropy during pre-training.

### A.5 CHOICE OF HYPER-PARAMETERS

During the experimental design, we wanted to ensure a fair comparison between the NCA and PNs. As a testimony of this effort, we obtained very competitive results for PNs (see for example the comparison to recent papers where architectures of similar capacity were used (Wang et al., 2019; Chen et al., 2019)). In particular:

- We always use the normalisation strategy of Wang et al. (2019), as it is beneficial also for PNs.
- Unless expressively specified, we always used PNs 5-shot model, which in our implementation outperforms the 1-shot model (for both 1-shot and 5-shot evaluation). Instead, (Snell et al., 2017) train and tests with the same number of shots.
- Apart from the episodes hyper-parameters of PNs, which we did search and optimise over to create the plots of Fig. 2, the only other hyper-parameters of PNs are those related to the training schedule, which are the same as the NCA. To set them, we started from the simple SGD schedule used by Wang et al. (2019) and only marginally modified it by increasing the number of training epochs to 120, increasing the batch size to 512 and setting weight decay and learning rate to $5e-4$ and $0.1$, respectively. As a sanity check, we trained both the NCA and PNs with the exact training schedule used by Wang et al. (2019). Results are reported in Table 6, and show that the schedule we used for this paper is considerably better for both PNs and NCA. In general, we observed that the modifications were beneficial for both NCA and PNs, and improvements in performance in NCA and PNs were highly correlated. This is to be expected given the high similarity between the two methods and losses.

### A.6 ADDITIONAL RESULTS FOR SEC. 4.2

Fig. 5 complements the results of Fig. 2 from Sec. 4.2

### A.7 ADDITIONAL RESULTS FOR SEC. 4.3

**Matching Networks (Vinyals et al., 2016)** are closely related to PNs (Snell et al., 2017), and even equivalent in the 1-shot case. The difference is in the use of the support set in the multi-shot case. Whereas PNs generate prototypes by averaging the embedding of the support set, Matching Networks adopt a weighted nearest-neighbour approach using an attention mechanism. If the attention mechanism is a softmax over the distances (which the authors suggest in Sec. 2.1.1 of their paper),

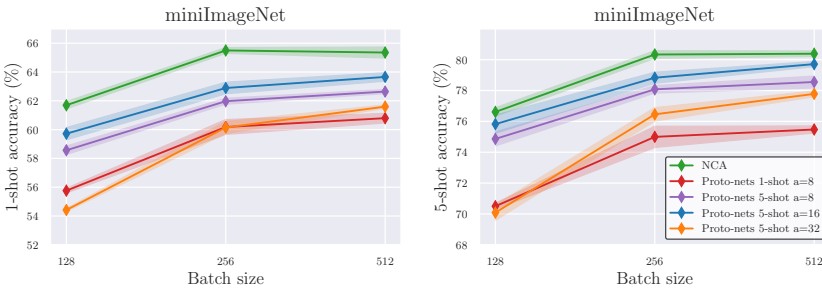

Figure 5: 1-shot (left) and 5-shot accuracies (right) on the validation set of *mini*ImageNet for different batch sizes. Models are trained using NCA or Proto-nets with different configurations: 1-shot with $a = 8$ and 5-shot with $a = 8$, 16 or 32. Reported values correspond to the mean accuracy of five models trained with different random seeds. Please see Sec. 4.2 for details.

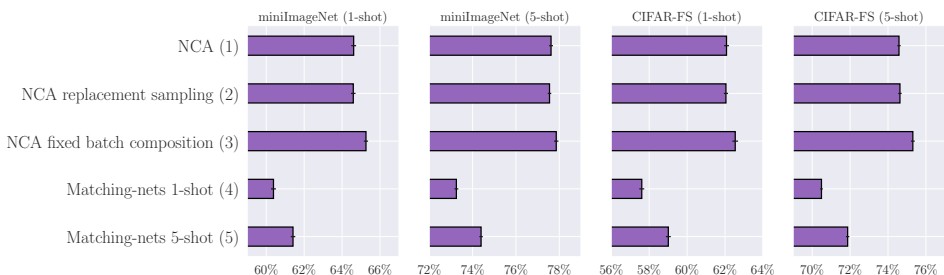

Figure 6: Ablation experiments on NCA and Matching Networks, both on batches or episodes of size 256 on the validation set of *mini*ImageNet and CIFAR-FS. All methods use *soft assignment* (Sec. 3.4) at test time.

we obtain the soft-assignment approach discussed in Sec. 3.4 of this paper. The only two differences between Matching Networks with a softmax attention mechanism and PNs are the lack of protoypes and the use of the cosine distance, instead of the Euclidean distance (Snell et al. (2017) has shown that the Euclidean distance is a better choice in FSL).

Given this similarity, and because of the relevance Matching Networks has in the few-shot learning community, we repeated the ablation experiment of Fig. 4. The results can be found in Table 6. In particular, we perform experiments on Matching Networks (without a Full Context Embedding) using the softmax attention mechanism, and using a Euclidean distance metric instead of a cosine distance metric. At training time, Matching Networks corresponds to the "no prototype" method in row 6 of Fig. 4. Therefore, the only difference between Matching Networks and NCA during training is the separation between the support and query set, leaving us with only one ablation to perform. At test time, evaluating Matching Networks is equivalent to using the soft-assignment approach described in Sec. 3.4. Therefore, for a fair comparison, for both NCA and "NCA fixed-batch composition" methods we also use the soft-assignment evaluation at test time.

As we can see, disregarding the separation between the support and query set also improves the performance of Matching Networks, and significantly so. This corroborates the findings of Sec. 4.3: the separation of roles between images in the support and query sets, typical of episodic learning, is detrimental to the performance of not only PN, but also Matching Networks. Instead, using the (closely related) NCA on standard random mini-batches allows for better exploitation of the training data, while simultaneously simplifying the training procedure.

**Ablation experiments for different batch sizes.** We repeated the ablation experiments done for batch size 256 (Fig. 4) also for size 128 and 512. Results can be found in Fig. 7 and Fig. 8. As we can see, the overall trend is maintained. A difference is the meaningful gap in performance between row 1 and 3 in Fig. 7 (size 128), which disppers in Fig. 8 (batch 512).

This is likely due to the number of positives available in an excessively small batch size. Since our vanilla NCA relies on using distance pairs and creates batches by simply sampling images randomly from the dataset, there is a limit to how small a batch can be (which depends on the number of classes

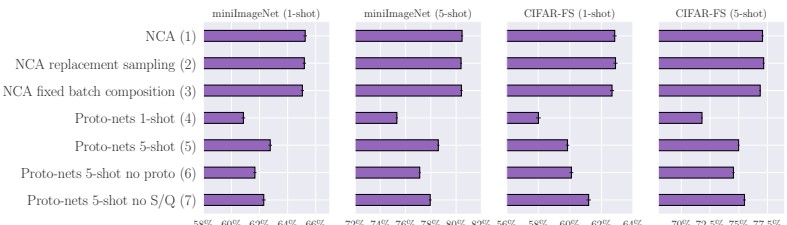

Figure 7: Ablation experiments on NCA and Prototypical Networks, both on batches or episodes of size 128 on the validation set of *mini*ImageNet and CIFAR-FS. Please refer to Sec. 4.3 for details.

Figure 8: Ablation experiments on NCA and Prototypical Networks, both on batches or episodes of size 512 on the validation set of *mini*ImageNet and CIFAR-FS. Please refer to Sec. 4.3 for details.

of the dataset). As an example, consider the extreme case of a batch of size 4. For the datasets considered, it is very likely that such a batch will contain no positive pairs. For a batch size of 128 and a training set of 64 classes, with a parameter-free sampler the NCA will have in expectation only one positive pair per class. Conversely, the NCA ablation with a fixed batch composition (i.e. with a set number of images per class) will have a higher number of positive pairs (at the cost of a reduced number of classes per batch). We believe this can explain the difference, as positive pairs constitute a less frequent (and potentially more informative) training signal. For the sake of simplicity, and since this only affects smaller batch sizes, we opted to use a vanilla, parameter-free sampler for the NCA in the rest of our experiments. Notice that, for batch size 512, there is even a slight (0.2%) decrease in performance using the fixed-batch composition w.r.t. the vanilla NCA.

## A.8 DETAILS ABOUT NUMBER OF PAIRS DESCRIPTION OF SECTION 3.3

In this section we demonstrate that the total number of training pairs that NCA can exploit within a batch is always strictly superior than the one exploited by the episodic batch strategy used by Prototypical Networks.

To ensure we have a "valid" episodic batch with a nonzero number of both positive and negative distance pairs, we assume that $n, m \geq 1$, and $w \geq 2$.

Below, we show that the number of positives for NCA, i.e. $\binom{m+n}{2}w$, is always greater or equal than the one for PNs, which is $mnw$:

$$
\begin{aligned}
\binom{m+n}{2}w &= \frac{(m+n)!}{2!(m+n-2)!}w \\
&= \frac{1}{2}(m+n)(m+n-1)w \\
&= \frac{1}{2}(m^2 + 2mn - m + n^2 - n)w \\
&= \frac{1}{2}(m(m-1) + 2mn + n(n-1))w \\
&\geq \frac{1}{2}(2mn)w = wmn.
\end{aligned}
$$

Similarly, we can show for negative distance pairs that $\binom{w}{2}(m+n)^2 > w(w-1)mn$:

| rank | method | # pos | # neg | # total pairs |
|---|---|---|---|---|
| 1 | *NCA* | 1792 | 129024 | 130816 |
| 2 | *5-shot a=16* | 1760 | 54560 | 56320 |
| 3 | *5-shot a=8* | 960 | 60480 | 61440 |
| 4 | *5-shot a=32* | 2160 | 32400 | 34560 |
| 5 | *1-shot a=8* | 448 | 28224 | 28672 |

Table 7: Number of positives and negatives used in the batch size 512 experiments of Fig. 2.

$$
\begin{aligned}
\binom{w}{2}(m+n)^2 &= \frac{w!}{2!(w-2)!}(m^2 + 2mn + n^2) \\
&= \frac{1}{2}w(w-1)(m^2 + 2mn + n^2) \\
&> \frac{1}{2}w(w-1)(2mn) \\
&= w(w-1)mn.
\end{aligned}
$$

This means that the NCA has at least the same number of positives as Prototypical Networks, and always has strictly more negative distances.

The total number of *extra* pairs that NCA can rely on is $\frac{w}{2}(w(m^2 + n^2) - m - n)$.

## A.9    DETAILS ABOUT NUMBER OF PAIRS DESCRIPTION OF SECTION 4.2

In Table 7 we plot the number of positives and negatives (gradients contributing to the loss) for the NCA and different episodic configurations of PNs, to see whether the difference in performance can be explained by the difference in the number of distance pairs that can be exploited in a certain batch configuration. This is often true, as the ranking can almost be fully explained by the number of total pairs in the right column. However, there are two exceptions to this: 5-shot with a=16 and 5-shot with a=8.

To understand this, we can see that the number of positive pairs is much higher for a=16 than for a=8. Since the positive pairs constitute a less frequent (and potentially more informative) training signal, this can explain the difference. The a=32 variant has an even higher number of positives than a=16, but the loss in performance there could be explained by a drastically lower number of negatives, and by the fact that the number of ways used during training is lower. So, while indeed generally speaking the higher number of pairs the better (which is also corroborated by Fig. 3, where moving right on the x-axis sees higher performance for both NCA and PNs), one should also consider how this interacts with the positive/negative balance and the number of classes present within a batch.

