# OpenReview forum: "On Episodes, Prototypical Networks, and Few-Shot Learning"
_ICLR.cc/2021/Conference — Reject_

### Official Review · AnonReviewer1 · 2020-10-24
**Review of On Episodes, Prototypical Networks, and Few-Shot Learning**

**Rating:** 5
**Confidence:** 4

**Review:**

Summary: This paper proposes to use neighborhood component analysis in lieu of prototype loss to train embedding functions of few-shot learning. This method takes full advantage of relations between all sampled points in an episode to facilitate learning, and it removes the distinction between support and query samples during meta-training time.

Reason for score: Overall, I lean towards reject. This paper proposes an interesting method that improves upon Prototypical Networks, and performs on-par with other baseline methods. However, the paper does not advance our understanding of how episodic training interacts with few-shot learning beyond the addition of more performance numbers to compare against.

Pros: The proposed method is a straight-forward improvement to Prototypical networks. In the wide range of scenarios evaluated in the experiments, the proposed method consistently outperforms ProtoNet. This proposed method should also be easy to implement, making integration with other FSL methods based on ProtoNets feasible.
While not exactly nouveau, using NCA to train embedding functions has not been done before (to my best knowledge) and is a theoretically sound approach.

Cons:
1. Perhaps unsurprisingly, the performance of NCA is lower than some FSL methods that use additional capacity. Even so, I think presenting only favorable comparisons in the performance tables is counter productive as it fails to capture the research context of this work.
2. The conclusion from the ablation studies on batch size is still unclear to me. The “NCA fixed batch composition” setting seems to perform better than NCA in 3 of the 4 plots in figure 4. This particular setting is interesting as it allows control of the number of classes in each batch. As the batchsize is fixed in the ablation, we won’t know what is the relation between the number of ways and performance of this fixed batch variant of NCA. This ablation is also confusing in that neither “no proto” nor “no S/Q” significantly improves performance, yet their combination performs well. A full ablation that systematically covers all hyperparameters would be helpful in furthering understanding in this direction.
3. The motivation of the paper feels unclear: on one hand the authors claim that they aim at understanding the (un)usefulness of episodic learning, yet on the other hand this paper doesn’t present any results beyond the final performance number to aid with this understanding. Visualizations and/or theoretical arguments would be greatly appreciated.

Minor points (suggestions, not related to score):
Introduction
“These results legitimately cast a doubt” -> “These results cast a doubt”
Related Work
“Between 2016 and 2017” feels unnecessary
“Matching and Prototypical…weighted by either an LSTM or a simple average, respectively”: Matching Networks also proposes a sample average variant. Their main difference lies in one uses cosine similarity while the other uses euclidean distance.
“Differently from these papers” -> Different
BACKGROUND AND METHOD
In 3.3, point 3 is not necessarily correct. Why would some examples be more likely than others in the episodic scheme? All FSL benchmarks (used in this paper) are close to class balanced, and an hierarchical sampling scheme for episodes results in full coverage each epoch just like standard supervised learning.
In 3.4 “which is the probability that image i is sampled from image j”. This is under the assumption that each support image defines a Gaussian in the embedding space. This is not true in general. The probabilistic interpretation of NCA should be further explained.

[Post rebuttal]
I am still leaning against the acceptance of this paper due to concerns about the limited impact of this submission. The topic of "limitations of episodic training" has been widely studied and in fact most SoA few-shot learning methods adopt a combination of supervised and episodic training for this precise reason.

The exploration into the relation between the number of sample pairs and learning performance is indeed correct, but no strong conclusions can be reached due to the logical jumps required. The authors established that the performance is correlated with number of pairs with a log/square-root curve, and that ProtoNet performance is similar to that of subsampled NCA (fig3). The mechanism behind why this is has not been elucidated. I think many questions can be explored to strengthen this paper, for example:
Are classes embedded tighter together?
Are hard negatives pushed further apart?
Does NCA induce a non-euclidean geometry that is more expressive than the euclidean geometry naturally induced by ProtoNets?
Can the classification problem be converted into a pair comparison problem, so that PAC learning theory can be used to explain the shape of this curve?
What is the sample complexity of the NCA classifier compared to the Prototypical classifier?

Regarding the proposed method, NCA is certainly an improvement over ProtoNets, but performs worse than existing methods in most experiments. This makes me doubtful of the impact of this work on the methodological front. Better modelling of relationships between few-shot examples has been implicitly and explicitly studied too. For instance, many works adopt sample level set functions in the form of transformers, attention modules, and graph neural networks. Arguably, these additional architectures are more expressive "deep" alternatives to NCA, and hence achieves better performance than the proposed method.

---

> ### Author Response · Authors · 2020-11-20
> **Answer to AnonReviewer1 - part 1/2**
>
> We thank this reviewer for their time, comments and suggestions.
>
> \> _“This paper proposes an interesting method [..] However, the paper does not advance our understanding of how episodic training interacts with few-shot learning beyond the addition of more performance numbers.”_
>
> \> _“This paper doesn’t present any results beyond the final performance number to aid with this understanding.”_
>
> Showing a difference in performance is surely part of our analysis (Fig.2, Table ~1~ 3 and ~2~ 4).
> However, we believe we go well beyond “the addition of more performance numbers”.
> We show why and how episodic learning in PNs is considerably data inefficient in several ways:
> * Intuitively, with a simple toy-case (the visualization of Fig.1) that shows how, by dividing the batch in support and query (sub-)set, many distances that would constitute useful training information are simply ignored.
> * By connecting the loss of PNs to the NCA in several steps (equations 2, 3, 4, 5 and 6)
> * By demonstrating that, by using episodes, PNs is not better than using the NCA loss while randomly discarding training examples from each batch (Fig.3).
>
> We would also like to point out to the reviewer a comparison against Matching Networks [Vinyals et al.] we reported in Appendix A.7, as suggested by AnonReviewer4 (the second reviewer). The results corroborate those obtained for Prototypical Networks in the main paper: the separation of roles between images in the support and query sets, typical of episodic learning, is detrimental to the performance of not only PNs, but also Matching Networks.
>
> \> _“Visualizations and/or theoretical arguments would be greatly appreciated.”_
>
> We provide the reader with several visualizations to illustrate our points (Fig.1 to Fig.4).
> Some of them are illustrative, others detail the experimental results.
> In particular, we would like to report a quote from AnonReviewer4: “I really like the way Figure 1 explains visually how Prototypical Networks miss out on useful relationships between examples in a batch and is therefore data-inefficient. To me, this is one of the submission’s most important contributions”
>
> \> _“unsurprisingly, the performance of NCA is lower than some FSL methods that use additional capacity [..] presenting only favorable comparisons [..] is counter productive”_.
>
> During the design of our experimental protocol, because of the large number of NCA/PNs configurations considered, we opted to perform experiments on the most popular architecture used in the evaluations of the recent literature, i.e. ResNet12.
> When choosing against which methods to compare in the final tables, given the very high number of papers on the topic, we clearly had to select a subset of them.
> We decided to select this subset by picking recent methods making use of episodic learning or presenting simple high-performing baselines based on pre-training with the cross-entropy loss.
> We do actually compare against methods that perform similarly or better than us. To make it more evident, we edited the tables by bolding the best method in each column, along with all other entries for which a 95% confidence interval test results in a non meaningful difference.
> In the paper update, we removed the multi-layer variant from the tables, as it is not very informative and it does not change the conclusions.
>
> Nonetheless, if we missed important comparisons please let us know which ones and we will update the tables accordingly.
>
> [continues below]

---

> > ### Author Response · Authors · 2020-11-20
> > **Answer to AnonReviewer1 - part 2/2**
> >
> > \> _“A full ablation that systematically covers all hyperparameters would be helpful”_
> >
> > \> _“The “NCA fixed batch composition” setting seems to perform better than NCA in 3 of the 4 plots in figure 4. This particular setting is interesting as it allows control of the number of classes in each batch.”_
> >
> > We agree that doing an ablation for each hyper-parameter would be interesting, but would also make the number of experiments explode, considering that we are running 5 seeds per configuration. To still address the spirit of this comment, we repeated the experiment of Fig.4 with different batch sizes.
> > Results are reported in Appendix A.7 and can help us explain the discrepancy between NCA and NCA with fixed batch composition (the second quoted question above).
> > The difference between “NCA with fixed-batch composition” and NCA is the highest with batch size 128 (1.3% on average), decreases with batch size 256 (0.5% on average) and becomes negligible with batch size 512 (NCA is actually 0.2% better).
> >
> > This is likely due to the number of positives available in an excessively small batch size.
> > Since our vanilla NCA relies on using distance pairs and creates batches by simply sampling images randomly from the dataset, there is a limit to how small a batch can be.
> > As an example, consider the extreme case of a batch of size 4.
> > For the datasets considered, it is very likely that such a batch will contain no positive pairs.
> > For a batch size of 128 and a training set of 64 classes, with a parameter-free sampler the NCA will have in expectation only one positive pair per class.
> > Conversely, the NCA ablation with a fixed batch composition (i.e. with a set number of images per class)  will have a higher number of positive pairs (at the cost of a reduced number of classes per batch).
> > We believe this can explain the difference, as positive pairs constitute a less frequent (and potentially more informative) training signal.
> > For the sake of simplicity, and since this only affects smaller batch sizes, we opted to use a vanilla, parameter-free sampler for the NCA in the rest of our experiments.
> > Notice that, for batch size 512, there is even a slight (0.2%) decrease in performance using the fixed-batch composition w.r.t. the vanilla NCA.
> >
> > We thank the reviewer for this suggestion and we believe that this is an interesting result. Understanding how to further improve NCA’s sampling strategy (a choice which is likely to be highly dependent on the distribution of classes within a dataset, besides the batch size) is an avenue for future work.
> >
> > \> _“This ablation [Fig4, row 6 and 7] is also confusing in that neither “no proto” nor “no S/Q” significantly improves performance, yet their combination performs well.”_
> >
> > We thank this reviewer for pointing it out.
> > We believe this phenomenon can be explained by counting the number of distance pairs contributing to the loss in each ablation. Using the formulas described at the end of Section 3.3, we compute the number of positives and negatives used for each ablation.
> > For row 6 there are 480 positives, and 14,880 negatives. For row 7 there are 576 positives and 20,170 negatives. In both cases, the number is significantly lower than the corresponding NCA, which gets 896 positives and 31,744 negatives, which is drastically more than either row 6 or 7 and likely explains the jump in performance that this reviewer pointed out.
> > Moreover, from row 5/6 to 7 we see a slight increase in performance, which can also be explained by the (slightly) larger number of distance pairs used in the loss.
> >
> >
> > \> [Sec 3.3] _“Point 3 is not necessarily correct. Why would some examples be more likely than others in the episodic scheme?”_
> >
> > We simply meant that, by sampling episodes with replacements, it is unlikely that all the images of the dataset will be visited. By doing a full pass of the dataset, instead, this is guaranteed. In any case, we did not observe this being a meaningful difference in our ablation studies.
> >
> > \> Thanks for the minor comments detailed at the end of the review - we edited the paper to address them.
> >
> > We hope that the above answers address the reviewer's comments - we are happy to discuss further.

---

> > > ### Comment · AnonReviewer1 · 2020-11-21
> > > **Thank you for the reply**
> > >
> > > Thank you for the additional experiments and explanations. I still have these following concerns and questions regarding the paper:
> > >
> > > I agree the the toy example presented in figure 1 is certainly interesting, but it is nonetheless a toy example. There is also the fact that as gradients are backpropagated through the centroid to each support embedding in ProtoNets, all pairs of support-query connections are still being used. While I don't claim that PN uses point-to-point connections more efficiently than NCA, this figure does not provide strong arguments against that claim. What I find missing are illustrative experiments that show how NCA's more efficient use of data improves the learned embedding function. Are support embeddings of different classes better separated? Or are embeddings of the same class more well grouped?
> > >
> > > Regarding experimental comparisons, FEAT (https://arxiv.org/pdf/1812.03664.pdf) and DeepEMD(https://arxiv.org/pdf/2003.06777.pdf) use episodic learning and report Resnet12 results, as two examples off the top of my head. I'd also note how most recent FSL papers report experimental results using several different backbone architectures to demonstrate the applicability of their method on various architectures, which I find lacking in this paper. This is an important point if the authors claim the methodological advances as an important contribution.
> > >
> > > Honestly, I am still confused on whether to judge this paper as a methodological paper or an "understanding" paper. On the methodological side, while NCA outperforms PN (a 3 years old method by now), many other methods also outperform PN and indeed NCA is not the best performing method even when controlled for architecture. The novelty factor here is also not exactly high since NCA is a well established method, albeit used in a different field. On the "understanding" side, I have to ask myself what new insights are gained about FSL from this paper. The fact that PN does not efficiently use relationships between all data points is a valid criticism, and this paper do support this claim to some extent. But, other works have also made similar observations and have come up with methods to address this problem. In your rebuttal, you quantified how many pairs of relationships are used in each ablation setting. These interesting observations could be connected and presented more coherently in the manuscript. Could a theoretical framework on sampling pairwise relationships and generalization be established to explain the issues of episodic learning? I feel that more theoretical explorations would greatly improve the impact of this work, but in its current scope, I think the potential impact of this work is still limited.

---

> > > > ### Author Response · Authors · 2020-11-24
> > > > **Additional answers to AnonReviewer1 - part 1/3**
> > > >
> > > > We thank the reviewer for their reply and further comments.
> > > >
> > > > \> _“I agree the toy example presented in figure 1 is certainly interesting, but it is nonetheless a toy example.”_
> > > >
> > > > \> _“While I don't claim that PN uses point-to-point connections more efficiently than NCA, this figure [Fig.1] does not provide strong arguments against that claim.”_
> > > >
> > > > At the end of Sec. 3.3, we specify the combinatorial formulas to exactly compute the number of positives and negatives in each setting, and show that the number of pairs in the non-episodic setting is strictly larger than in the episodic setting. The toy example is useful to visually outline the discrepancy between the two setups, but the observations can be generalised to any batch composition.
> > > > To make this more clear, we now report the total number of pairs contributing to the loss in Table 1. Moreover, in Appendix A.8 we derive how the total number of pairs used in the loss of NCA is strictly greater than the one used in the loss of PNs, no matter the batch size and the hyper-parameters used for the episode composition, and how that difference grows exponentially, and as $O(w^2(m^2+n^2)$.
> > > > More precisely, the number of _extra_ pairs that the NCA loss can utilise from a batch is $\frac{1}{2}w(w(m^2 + n^2) - m - n)$.
> > > > Using for example the episodic batch composition used by [Snell et al.] for their 5-shot model training (w=20, m=15, n=5), we can see how this difference is important (about 50k unused pairs for a batch of size 400).
> > > >
> > > > We hope that this extension to the combinatorial analysis can show more immediately the important difference in batch exploitation between the methods considered, and we thank the reviewer for having pointed in this direction with their comments.
> > > >
> > > > One could still posit that it is possible that, for PNs, the way the pairs are sampled within a batch is beneficial despite not all of them are exploited, e.g. by somehow mimicking the episode composition seen at test time.
> > > > In the second experiment of Sec. 4.2 (from the paragraph now titled “Episodic batches vs. random sub-sampling of standard batches) we demonstrated how this is actually not the case. We show that even if we sample only a _subset_ of the distances used in the NCA loss computation, we still get better - and at most equal - performance compared to episodic setups for prototypical networks which use an equivalent number of distance pairs.
> > > >
> > > > This result shows that, in PNs, there is nothing inherently special about the episodic strategy, and it is at best equivalent to randomly discarding pairs within the NCA framework.
> > > > This experiment thus shows that PNs do not use point-to-point connections more efficiently than NCA, and that it is beneficial to fully exploit the data within a batch.
> > > >
> > > > \> _“There is also the fact that as gradients are back propagated through the centroid to each support embedding in ProtoNets, all pairs of support-query connections are still being used.”_
> > > >
> > > > This is correct, and all the “pairs” that we have counted take this into account already, which includes the previous answer in this thread, and also in the answer to AnonReviewer4 (the second reviewer). Notice that we stated in the original submission, at the end of Sec.3.3: _“When considering the number of distances contributing to the loss, points within the support  set  count  independently  and  thus  we  get  to wmn= 9 positives  and w(w−1)mn=18 negatives.”_
> > > > Whereas, if we only considered distances with centroids, the number of positives and negatives would be 3 and 6, respectively.
> > > > Given your remark we have now made this more clear in the paper.
> > > >
> > > > \> _“What I find missing are illustrative experiments that show how NCA's more efficient use of data improves the learned embedding function. Are support embeddings of different classes better separated? Or are embeddings of the same class more well grouped?”_
> > > >
> > > > Note that the evaluation framework used in all our experiments already implies this, because the inductive bias on which nearest-neighbour methods  are based is that points with the same labels are closer together in the embedding space.
> > > > As such, better FSL performance does indeed translate to better class separateness between the learned embeddings.
> > > >
> > > > [continues below]

---

> > > > > ### Author Response · Authors · 2020-11-24
> > > > > **Additional answers to AnonReviewer1 - part 2/3**
> > > > >
> > > > > \> _“Honestly, I am still confused on whether to judge this paper as a methodological paper or an understanding paper.”_
> > > > > Using the terminology suggested by this reviewer, our paper mostly falls into the “understanding” category.
> > > > >
> > > > > Our goal was not to develop a new SOTA method.
> > > > > Instead, motivated by the recent success of simple cross-entropy-based methods in FSL,  we wanted to investigate whether episodic learning is always a correct choice, and we started with a case study on Prototypical Networks, which is arguably one of the most popular algorithms and has largely influenced the community (it has 1700 citations, and many methods are directly based on them).
> > > > >
> > > > > Our paper draws the connection between Prototypical Networks and NCA, where NCA can be seen as the non-episodic variant of PNs (and of Matching Network too, as shown in the experiments of Appendix A.7).
> > > > > With our analysis, we show that the fundamental problem of the episodic strategy in PNs is the poor data efficiency.
> > > > >
> > > > > Nonetheless, we believe our paper also provides a minor practical contribution, as the vanilla NCA makes a simple and appealing baseline for future work.
> > > > >
> > > > > \> _“Other works have also made similar observations and have come up with methods to address this problem. “_
> > > > >
> > > > > To the best of our knowledge, the analysis we did on Prototypical and Matching Networks is novel, and we kindly invite the reviewer to specify which works they are referring to.
> > > > > There are papers that show how non-episodic, cross-entropy-based methods can perform well, which we cited in the paper since the Introduction (e.g. [Chen et al.; Wang et al.; Dhillon et al.; Tian et al.]. However, that is still considerably different from proposing a case study which illustrates where the shortcomings of a previous method are, and how to overcome them.
> > > > > Importantly, we believe that the insight we provide is important because it puts into discussion what was thought to be the edge brought by episodic learning, as it can be gathered by these quotes:
> > > > > > [Snell et al., 2017] _“The use of episodes makes the training problem more faithful to the test environment and thereby improves generalization.”_
> > > > >
> > > > >  > [Vinyals et al., 2016]: _“[..] our training procedure is based on a simple machine learning principle: test and train conditions must match. Thus [we] train our network to do rapid learning [..], much like how it will be tested when presented with a few examples of a new task.”_
> > > > >
> > > > > Instead, for Prototypical Networks and Matching Networks this episodic structure is at the core of a serious data-inefficiency problem, which actually negatively affects the performance.
> > > > >
> > > > > As we mentioned in the paper, the use-case of [Raghu et al.] is the paper most similar to ours in spirit. They show that MAML does not change its internal representation very rapidly, but rather learns features during training which are reused at test time.
> > > > > We believe that our paper, [Raghu et al.] and the several cross-entropy baselines mentioned above can be considered, together, as an indication for the community that episodic learning should be considered with caution when designing a new FSL method.
> > > > >
> > > > > \> _“Most recent FSL papers report experimental results using several backbone architectures [..] This is an important point if the authors claim the methodological advances as an important contribution.”_
> > > > >
> > > > > As discussed in the points above, the aim of our paper is to understand a phenomenon, and not to beat the state-of-the-art. For this reason, we dedicated our computational budget mainly to the many experiments of Sec. 4.1 to 4.3.
> > > > >
> > > > > We can see that in Fig.2 we trained 16 different setups (there were more for 1-shot PNs that we did not plot since they performed significantly worse), for both CIFAR-FS and miniImageNet (Fig. 5 in Appendix A.7).  Moreover, each number that we report is the average accuracy over 50k episodes, where we trained 5 models with 5 different seeds and evaluated them on 10k episodes for each seed, which has been particularly appreciated by another reviewer.
> > > > > _“[..] the authors went beyond the usual practice of reporting accuracies on a single run and instead trained each method with five different random initializations, and this is a practice that I’m happy to see in a few-shot classification paper”._
> > > > >
> > > > > This means that for Fig. 2 and 5, we already had to train 120 models - excluding any hyper-parameter search and the episodic batch setups we did not report.
> > > > > Note that this suite of experiments has been helpful in determining the best set of episodic hyper-parameters to use for the PNs implementation we compare against in the two tables of Sec. 4.4.
> > > > >
> > > > > [continues below]

---

> > > > > > ### Author Response · Authors · 2020-11-24
> > > > > > **Additional answers to AnonReviewer1 - part 3/3**
> > > > > >
> > > > > > We thought that repeating this for other architectures was unfeasible from a computational budget perspective, and thus we opted to do an as thorough as possible analysis on the most popular architecture.
> > > > > >
> > > > > > \> _“FEAT and DeepEMD use episodic learning and report Resnet12 results”_
> > > > > >
> > > > > > Given the number of papers on the topic (about 5 a week in the last year alone), the comparison with the state-of-the-art of Sec. 4.4 cannot possibly be exhaustive.
> > > > > > Instead, it is intended to be a way to contextualise the vanilla NCA performance with respect to other modern _simple_ methods. Since we do not claim to improve state-of-the-art but only to be competitive, and given that our contributions revolve around furthering the understanding of previous methods and not around the proposal of a new one, we thought this would have been sufficient.
> > > > > > We slightly edited the text in Sec. 4.4 to make our criteria clearer.
> > > > > >
> > > > > > FEAT and DeepEMD are two interesting papers that came out this year and test on ResNet12 backbone. We did not include them in our comparison for the same reason we did not include the self-distillation variant of [Tian et al.]: they present a fairly complicated setup (in particular, they both perform adaptation at test time) with insights that are likely to be applicable also to our method.
> > > > > > More specifically, the FEAT paper experiments with different set-to-set strategies to adapt the model learned on seen classes to the new classes presented at test time. From Figure 2b in their paper, it should be clear how their contribution is in the adaptation layer, which could be used in our implementation as well. Moreover, the introduction of these set-to-set functions requires additional capacity in the form of several FC layers (see Fig.5 of their paper).
> > > > > > DeepEMD shows the benefit of working with local information (as opposed to image-level embeddings) in a metric-learning FSL framework.
> > > > > > To do so, the average pooling layer is removed and individual images are considered as a collection of local embeddings rather than individual representations.
> > > > > > This is an important insight, but it is likely to provide improvement for different methods too, including the vanilla NCA.
> > > > > > Moreover, also here during evaluation part of the model is adapted at test time.
> > > > > > As an aside, note that since DeepEMD formulation expressly does not require a functional separation between query and support images (see Fig.2 of their paper), the conclusions on data-inefficiency of PNs we draw in our paper might also benefit this method (although of course this needs to be proven empirically)
> > > > > >
> > > > > > \> _“in your rebuttal, you quantified how many pairs of relationships are used in each ablation setting. These interesting observations could be connected and presented more coherently in the manuscript.”_
> > > > > >
> > > > > > We appreciate the suggestion to improve the clarity of the discussion.
> > > > > > We have included in the ablation study (Sec. 4.3) a discussion on the number of pairs visited in each variant.
> > > > > > Furthermore, in Appendix A.9 we have also included a discussion on the effect of the number of pairs for the experiment of Fig. 2, to which we refer in the answer to AnonReviewer4 (the second reviewer).
> > > > > >
> > > > > > We relate these new observations to our extended discussion about the number of pairs at the end of Sec. 3.3.
> > > > > >
> > > > > > \> _“Could a theoretical framework on sampling pairwise relationships and generalization be established to explain the issues of episodic learning.”_
> > > > > >
> > > > > > We are not entirely sure what it is meant here with “theoretical framework”.
> > > > > > If the request is to outline more precisely the extent of the data-inefficiency brought by episodic sampling in PNs, we hope that the extension of the combinatorial analysis we added in the latest update is satisfactory.
> > > > > >
> > > > > > Another interpretation is that this reviewer would appreciate a framework that is capable of generalising the conclusions that can be drawn from our work to any method that samples batches with episodic learning.
> > > > > > This is surely interesting, but we believe it should be considered as future work that can be inspired by our results, and not as something to be already expected as a contribution to our paper.
> > > > > >
> > > > > > Moreover, an all-encompassing framework that covers generic meta-learning algorithms could result in a very challenging endeavor, unless strong simplifications are introduced.
> > > > > > Unlike Prototypical Networks (and the simple version of Matching Networks that we considered), for many meta-learning algorithms the support set and the query set have different functional roles, i.e. training and testing a base-learner within the meta-learning inner loop (see [Hospedales et al.]).
> > > > > > As such, the sampling-inefficiency component outlined in this paper will be inevitably confounded with an algorithmic design that is hard to ablate without significantly altering its formulation, thus making it hard to design conclusive experiments.
> > > > > >
> > > > > > We thank again this reviewer for their time and valuable insights. We hope that our answers address the concerns raised.

---

### Official Review · AnonReviewer3 · 2020-10-28
**Technical contributions are not clear and experiments are generic**

**Rating:** 5
**Confidence:** 3

**Review:**

This paper investigates the usefulness of episodic learning in prototypical learning which is a popular practice in few-shot learning. The authors propose a non-episodic prototypical network which basically corresponds to the classical neighborhood component analysis and they claimed that this network reliably improves over its episodic counterpart in multiple datasets. I have the following comments on the paper:

1. The sections 3.1, 3.2 and 3.3 are not the contributions of the paper. Only section 3.4 can be considered as something new from experimental point of view and not methodologically new. k-NN, 1-NN with class centroids and soft assignments are all some specific experimental settings. Therefore, I do not see the technical contributions of the paper other than the claimed novel experimental settings which is also marginal.

2. The paper shows a robust experimentations and comparisons with prior arts, however, I don't understand how the three settings mentioned in the section 3.4 are evaluated in the tables.

3. I am curious if you have done a comparison with baseline NCA, i.e. equation (3). I have not found the comparison in A.1 which only contains some discussions but no direct comparison.

4. Anyways, The paper is written in good English and I haven't found any typos yet.

Based on my current understandings and above comments, currently I recommend for a weak rejection. However, I would like to follow the discussions on the paper and understand the contributions well.

---

> ### Author Response · Authors · 2020-11-20
> **Answer to AnonReviewer3 - part 1/2**
>
> We thank this reviewer for their comments and their interest in following the discussion. The review contains three comments, that we address in order below:
>
> \> _“The sections 3.1, 3.2 and 3.3 are not the contributions of the paper [..] I do not see the technical contributions of the paper other than the claimed novel experimental settings”_
>
> We titled Section 3 “Background and method” because it was important to give details on episodic learning, Prototypical Networks and NCA in order to introduce the readers to our contributions, which are mainly in Section 4, the experimental section.
> In particular, our contributions are:
>  * We highlighted the connection between NCA and PNs, and demonstrated that for PNs “episodes” are a data-inefficient way of exploiting the training signal available in a batch.
>  * We showed empirically that, for PNs, episodic learning achieves an analogous performance to discarding distance pairs, at random, from a standard mini-batch using the NCA (Fig.3).
> * We showed that training this vanilla NCA loss, an extremely simple method with almost no hyper-parameters, leads to performance that is competitive with the state-of-the-art.
>
> These results are novel and, given the relevance of Prototypical Networks (~1700 citations) and the great amount of work it has influenced, should be of significant interest for this community.
>
> We would like to highlight some of AnonReviewer4’s (the 2nd reviewer) comments, which highlight the contributions of this paper from an external perspective: _“The value of episodic training is increasingly being questioned, and the submission approaches the topic from a new and interesting perspective.”_; and
> _“The connection between nearest-centroid few-shot learning approaches and NCA has not been made in the literature to my knowledge and has potential applications beyond the scope of this paper.”_
>
> Regarding the lack of technical contribution mentioned by this reviewer: ICLR’s guidelines stress on the importance of novel findings, which applies to our case.
> For instance, “Understanding deep learning requires rethinking generalization” [Zhang et al.] has been nominated as one of the three best papers from 2017’s edition of ICLR and does not introduce anything new, algorithmically; its contributions revolve around the experimental findings.
> Clearly, we do not want to qualitatively compare ourselves to that paper, we just wanted to show that it is possible for a paper to be accepted (and sometimes to thrive) mainly for the novel insights obtained by the experimental analysis.
>
> \> _“The paper shows a robust experimentations and comparisons with prior arts, however, I don't understand how the three settings mentioned in the section 3.4 are evaluated.”_
>
> Apologies if this was not clear. We simply tried all the variants to perform classification on the same model trained with the NCA, and we picked the 1-NN with centroids for the rest of the experiments as it was the best.
> We discussed this in the third paragraph of Section 4.1: “We compared the inference methods discussed in Sec. 3.4 on miniImageNet and CIFAR-FS. Results can be found in Table ~3~ 2.  We chose to use 1-NN with class centroids in all our experiments, as it performs significantly better than k-NN or Soft Assignment.”
>
> Table ~3~ 2 used to be in the Appendix, but for clarity we moved it in the main text, close to where it is referenced.
>
> [continues below]

---

> > ### Author Response · Authors · 2020-11-20
> > **Answer to AnonReviewer3 - part 2/2**
> >
> > \> _“I am curious if you have done a comparison with baseline NCA, i.e. equation (3). I have not found the comparison in A.1 which only contains some discussions but no direct comparison.”_
> >
> > We used a vanilla NCA throughout the experiments.
> > * The NCA loss (eq. 3) is used to train all the methods that are referred to as NCA in the plots and tables of Section 4. The NCA is optimised during training and there is no test-time adaptation (except for one experiment in Appendix A.1).
> > * Section A.1 from the Appendix seeks to understand whether adapting to the support set provided during the evaluation by learning a positive semi-definite matrix is beneficial. Results are actually provided, in Table 5 in the Appendix, and show only small improvements. As we said in A.1: “given the computational cost, we opted for non performing adaptation to the support sets in our experiments of Sec. 4 [the experimental section of the paper]”.
> >
> > Does this answer this reviewer’s question? We were unsure how to interpret this comment.
> >
> > We would also like to point out to the reviewer a comparison against Matching Networks [Vinyals et al.] we reported in Appendix A.7. The results corroborate those obtained for Prototypical Networks in the main paper: the separation of roles between images in the support and query sets, typical of episodic learning, is detrimental to the performance of not only PNs, but also Matching Networks.
> >
> > \> _“I would like to follow the discussions on the paper and understand the contributions well.”_
> >
> > We hope that our answers have addressed this reviewer’s comments about our submission.
> > If not, we are happy to continue the conversation.
> >
> > * [Vinyals et al.] Matching Networks for One Shot Learning - NeurIPS 2016
> > * [Zhang et al.]Understanding deep learning requires rethinking generalization - ICLR 2017

---

### Official Review · AnonReviewer4 · 2020-10-28

**Rating:** 7
**Confidence:** 5

**Review:**

#### Summary

The submission investigates the properties of episodic training and its impact on learning using Prototypical Networks as a case study. The paper draws a connection between Prototypical Networks and Neighbourhood Component Analysis (NCA), noting that their loss functions are similar but that NCA is trained non-episodically, which allows it to learn from the relationship between all example pairs in a batch.

When controlling for batch size, the paper claims to show that NCA (combined with a nearest-centroid inference strategy) performs better than Prototypical Networks, as evidenced by experiments on CIFAR-FS and mini-ImageNet. Ablation experiments are performed, claiming to show that applying the NCA loss to batches sampled episodically allows Prototypical Networks to bridge the performance gap with NCA, and that the partition of examples within a batch into support and query sets is detrimental to Prototypical Networks training. Finally, NCA is evaluated alongside comparable competing approaches on mini-ImageNet, CIFAR-FS, and tiered-ImageNet, and is claimed to yield results comparable or superior to the state-of-the-art.

#### Strengths and weaknesses

* **+** The value of episodic training is increasingly being questioned, and the submission approaches the topic from a new and interesting perspective.
* **+** The connection between nearest-centroid few-shot learning approaches and NCA has not been made in the literature to my knowledge and has potential applications beyond the scope of this paper.
* **+** The paper is well-written, easy to follow, and well-connected to the existing literature.
* **-** The extent to which the observations presented generalize to few-shot learners beyond Prototypical Networks is not evaluated, which may limit the scope of the submission’s contributions in terms of understanding the properties of episodic training.
* **-** The Matching Networks / NCA connection makes more sense in my opinion than the Prototypical Networks / NCA connection.
* **-** A single set of hyperparameters was used across learners for a given benchmark, which can bias the conclusions drawn from the experiments.

#### Recommendation

I’m leaning towards acceptance. I have some issues with the submission that are detailed below, but overall the paper presents an interesting take on a topic that’s currently very relevant to the few-shot learning community, and I feel that the value it brings to the conversation is sufficient to overcome the concerns I have.

#### Detailed justification

The biggest concern I have with the submission is methodological. One one hand, the authors went beyond the usual practice of reporting accuracies on a single run and instead trained each method with five different random initializations, and this is a practice that I’m happy to see in a few-shot classification paper. On the other hand, the choice to share a single set of hyperparameters across learners for a given benchmark leaves a blind spot in the evaluation. What if Prototypical Networks are more sensitive to the choice of optimizer, learning rate schedule, and weight decay coefficient than NCA? Is it possible that the set of hyperparameters chosen for the experiments happens to work poorly for Prototypical Networks? Would we observe the same trends if we tuned hyperparameters independently for each experimental setting? In its current form the submission shows that Prototypical Networks are sensitive to the hyperparameters used to sample episodes *while keeping other hyperparameters fixed*, but showing the same trend while doing a reasonable effort at tuning other hyperparameters would make for a more convincing argument. This is why I take the claim made in Section 4.2 that "NCA performs better than all PN configurations, no matter the batch size" with a grain of salt, for instance.

I also feel that the submission misses out on an opportunity to support a more general statement about episodic training via observations on approaches such as Matching Networks, MAML, etc. I really like the way Figure 1 explains visually how Prototypical Networks miss out on useful relationships between examples in a batch and is therefore data-inefficient. To me, this is one of the submission’s most important contributions: the suggestion that a leave-one-out strategy could allow episodic approaches to achieve the same kind of data efficiency as non-episodic approaches, alleviating the need for a supervised pre-training / episodic fine-tuning strategy. To be clear, I don’t think the missed opportunity would be a reason to reject the paper, but I think that showing empirically that the leave-one-out strategy applies beyond Prototypical Networks would make me lean more strongly towards acceptance.

The connection drawn between Prototypical Networks and NCA feels forced at times. In the introduction the paper claims to "show that, without episodic learning, Prototypical Networks correspond to the classic Neighbourhood Component Analysis", but Section 3.3 lists the creation of prototypes as a key difference between the two which is not resolved by training non-episodically. From my perspective, NCA would be more akin to the non-episodic counterpart to Matching Networks without Full Contextual Embeddings – albeit with a Euclidean metric rather than a cosine similarity metric – since both perform comparisons on example pairs.

This relationship with Matching Networks could be exploited to improve clarity. For instance, row 6 of Figure 4 can be interpreted as a Matching Networks implementation with a Euclidean distance metric. With this in mind, could the difference in performance between "*1*-NN with class centroids" and *k*-NN / Soft Assignment noted in Section 4.1 – as well as the drop in performance observed in Figure 4’s row 6 – be explained by the fact that a (soft) nearest-neighbour approach is more sensitive to outliers?

Finally, I have some issues with how results are reported in Tables 1 and 2. Firstly, we don’t know how competing approaches would perform if we applied the paper’s proposed multi-layer concatenation trick, and the idea itself feels more like a way to give NCA’s performance a small boost and bring it into SOTA-like territory. Comparing NCA without multi-layer against other approaches is therefore more interesting to me. Secondly, 95% confidence intervals are provided, but the absence of identification of the best-performing approach(es) in each setting makes it hard to draw high-level conclusions at a glance. I would suggest bolding the best accuracy in each column along with all other entries for which a 95% confidence interval test on the difference between the means is inconclusive in determining that the difference is significant.

#### Questions

1. In Equation 2, why is the sum normalized by the total number of examples in the episode rather than the number of query examples?
1. Can the authors comment on the extent to which Figure 2 supports the hypothesis that NCA is better for training because it learns from a larger number of positives and negatives? Assuming this is true, we should see that Prototypical Networks configurations that increase the number of positives and negatives should perform better for a given batch size. Does Figure 2 support this assertion?
1. Can the authors elaborate on the "no S/Q" ablation (Figure 4, row 7)? What is the point of reference when computing distances for support and query examples? Is the loss computed in the same way for support and query examples? The text in Section 4.3 makes it appear like the loss for query examples is the NCA loss, but the loss for support examples is the prototypical loss. Wouldn’t it be conceptually cleaner to compute leave-one-out prototypes, i.e. leave each example out of the computation of its own class’ prototype (resulting in slightly different prototypes for examples of the same class)? In my mind, this would be the best way to remove the support/query partition while maintaining prototype computation, thereby showing that the partition is detrimental to Prototypical Networks training.

#### Additional feedback

1. This is somewhat inconsequential, but across all implementations of episodic training that I have examined I haven’t encountered an implementation that uses a flag to differentiate between support and query examples. Instead, the implementations I have examined explicitly represent support and query examples as separate tensors. I was therefore surprised to read that "in most implementations [...] each image is characterised by a flag indicating whether it corresponds to the support or the query set [...]"; can the authors point to the implementations they have in mind when making that assertion?
1. I would be careful with the assertion that "during evaluation the triplet {w, n, m} [...] must stay unchanged across methods". While this is true for the benchmarks considered in this submission, benchmarks like Meta-Dataset evaluate on variable-ways and variable-shots episodes.
1. I’m not too concerned with the computational efficiency of NCA. The pairwise Euclidean distances can be computed efficiently using the inner- and outer-product of the batch of embeddings with itself.

---

> ### Author Response · Authors · 2020-11-20
> **Answer to AnonReviewer4 - part 1/3**
>
> We thank this reviewer for their time on writing a very detailed and insightful review.
>
> \> _“NCA would be more akin to the non-episodic counterpart of Matching Networks without Full Contextual Embeddings – albeit with a Euclidean metric rather than a cosine similarity metric – since both perform comparisons on example pairs.”_
>
> \> _“Showing empirically that the leave-one-out strategy applies beyond Prototypical Networks would make me lean more strongly towards acceptance”_
>
> We initially considered including Matching Networks, but eventually we decided against it for practical reasons:
> * The original paper proposes multiple variants.
> * The original paper uses cosine similarity, which has been subsequently shown to be a poor choice in the FSL setting.
> * To the best of our knowledge, an official implementation of Matching Networks does not exist.
>
> However, we agree that Matching Networks are very relevant for us, and we thank this reviewer for specifying a setting to extend our analysis.
> We followed the suggestion and repeated the ablation analysis of Fig.4 to Matching Networks without contextual embeddings and with a Euclidean metric.
> Results are illustrated in Section A.7, and corroborate those coming from PNs analysis: the separation of roles between images in the support and query sets is also significantly detrimental to the performance of Matching Networks
>
> \> _“The choice to share a single set of hyperparameters [..] leaves a blind spot in the evaluation”_.
>
> During the experimental design, we believe we dedicated a significant effort in ensuring apple-to-apple comparisons against a very competitive implementation of PNs.
> As a testimony of this effort, the results achieved by our implementation of PNs are very competitive (see for example the comparison to recent papers where architectures of similar capacity were used [Wang et al.], [Chen et al.])
> However, in our submission we did not do a great job in explaining our choices.
>
> In general, every time we did something that departed from the original PNs implementation, we verified that this was beneficial also for PNs. In particular:
> * We always use the normalisation adopted in SimpleShot [Wang et al.], which is beneficial also for PNs.
> * In the comparison tables of Sec.4, we always used PNs’ 5-shot model, which in our implementation always outperforms the 1-shot model (for both 1- and 5-shot). Instead, [Snell et al.] train and test with the same number of shots.
> * Apart from the episodes hyper-parameters of PNs, which we did search and optimise over to create the plots of Fig.2, the only other hyper-parameters of PNs are those related to the training schedule, which are the same as NCA.
> To set them, we started from the simple SGD schedule used by SimpleShot [Wang et al.] and only marginally modified it by increasing the number of training epochs to 120, increasing the batch size to 512 and setting weight decay and learning rate to 5e-4 and 0.1 respectively. We observed that these changes were beneficial for both NCA and PNs.
>
> A few points to show empirically that the small set of choices we made for the hyper-parameters are beneficial for both NCA and PNs and that we conducted fair comparisons:
>
> a) As a sanity check, we tried to train PNs with the learning schedule used in [Snell et al] for both miniImageNet and CIFAR-FS, and we observed consistently inferior performance with respect to what we obtained with ours (between -1% and -2.5% depending on the dataset).
>
> b) We trained both the NCA and PNs with the training schedule used in SimpleShot [Wang et al., 2019]. Detailed results are reported in Table ~5~ 6 in the Appendix. The schedule we used in the paper is considerably better for both PNs and NCA.
>
> c) Finally, we have run a small 4x3 grid search for the learning rate and weight decay hyper-parameters. For both NCA and PNs, we used the best setup from Fig.2.
> Results on the validation set of CIFAR-FS are reported in the ASCII tables below.
>
> Performance refers to testing on 1-shot / 5-shot. Confidence intervals are omitted for reason of space, but are between 0.15 and 0.20.
>
> [continues below]

---

> > ### Author Response · Authors · 2020-11-20
> > **Answer to AnonReviewer4 - part 2/3**
> >
> > [PNs with a=16, 5-shots, batch-size=512]
> > ~~~
> > +-----------+---------------+---------------+---------------+
> > |       	 |   wd = 1e-3   |   wd = 5e-4   |   wd = 1e-4  |
> > +-----------+---------------+---------------+---------------+
> > | lr = 0.5  | 44.61 / 57.66 | 56.12 / 70.95 | 59.01 / 74.31 |
> > | lr = 0.1  | 59.89 / 74.94 | 60.46 / 75.53 | 58.97 / 73.75 |
> > | lr = 0.05 | 61.08 / 76.02 | 60.91 / 75.79 | 59.28 / 74.24 |
> > | lr = 0.01 | 59.78 / 74.64 | 58.82 / 73.85 | 58.34 / 73.22 |
> > +-----------+---------------+---------------+---------------+
> > ~~~
> >
> > [NCA, batch-size=512]
> > ~~~
> > +-----------+---------------+---------------+---------------+
> > |       	 |   wd = 1e-3   |   wd = 5e-4   |   wd = 1e-4  |
> > +-----------+---------------+---------------+---------------+
> > | lr = 0.5  | 50.28 / 64.31 | 58.24 / 72.83 | 61.23 / 75.72 |
> > | lr = 0.1  | 62.52 / 76.97 | 62.87 / 77.10 | 61.45 / 75.70 |
> > | lr = 0.05 | 63.49 / 77.55 | 62.77 / 76.90 | 61.29 / 75.58 |
> > | lr = 0.01 | 61.70 / 75.92 | 60.85 / 75.08 | 60.04 / 74.32 |
> > +-----------+---------------+---------------+---------------+
> > ~~~
> >
> > Notice how the performance of NCA and PNs are highly correlated across the board, and it is not the case that PNs are more sensitive to the learning rate or weight decay. For both PNs and NCA, the maximiser is actually at {wd=1e-3, lr=0.05} and achieves about 0.5% better performance than what we used (i.e. wd=5e-4, lr=0.1); however, this is to be expected because we chose the hyper-parameters early in the project. As a matter of fact, our focus initially was not the comparison between NCA and PNs, and we were running exploratory experiments on both as baselines.
> >
> > \> [Re. the] _“multi-layer concatenation trick, idea feels more like a way to give NCA’s performance a small boost. Comparing NCA without multi-layer against other approaches is therefore more interesting”_.
> >
> > We agree the focus should be on the vanilla NCA, and that concatenating features from different layers might turn to be an advantage for other methods too.
> > We removed the multi-layer variant from the main tables and deferred the description of its effects to Appendix A.1.
> > All the considerations made in Sec. 4.4 still apply: vanilla NCA can still be considered competitive, and sometimes (on CIFAR-FS) superior, to the state-of-the-art.
> >
> >
> > \> _“I would suggest bolding the best accuracy in each column”_.
> >
> > We agree this improves legibility - we edited the tables accordingly.
> >
> > \> _“could the difference in performance between "1-NN with class centroids" and k-NN / Soft Assignment [..] be explained by the fact that a (soft) nearest-neighbour approach is more sensitive to outliers?”_
> >
> > We agree with the reviewer that this is the most likely explanation.
> > Since the classes between training and evaluation are disjoint, the model is unlikely to produce calibrated probabilities.
> > As such, within the softmax, outliers behaving as false positives can happen to highly influence the final decision, and those behaving as false negatives can end up being almost completely ignored (their contribution is squashed to zero).
> > With the nearest centroid classification approach outliers are still clearly an issue, but their effect can be less dramatic.
> >
> > **Questions**
> >
> > \> _“In Equation 2, why is the sum normalized by the total number of examples in the episode rather than the number of query examples?”_
> >
> > Thanks for pointing this out. This is a typo and has now been corrected.
> >
> > \> _“Can the authors comment on the extent to which Figure 2 supports the hypothesis that NCA is better for training because it learns from a larger number of positives and negatives? Assuming this is true, we should see that PNs configurations that increase the number of positives and negatives should perform better for a given batch size. Does Figure 2 support this assertion?”_
> >
> > This is a good observation, and we believe that Fig. 2 does indeed support the hypothesis, with some minor caveats that we now discuss. Below, we plot a table outlining the number of positives and negatives (gradients contributing to the loss) for the NCA and different episodic configurations of PNs. We consider the case where the batch size is 512.
> > ~~~
> > +------+-----------------+-------+--------+-----------------+
> > | rank |   method	    | # pos | # neg  | # total pairs  |
> > +------+------------------+-------+--------+----------------+
> > |	0 | NCA     	    |  1792 | 129024 |    	    130816 |
> > |	1 | PNs 5-shot a=16 |  1760 |  54560 |     	    56320   |
> > |	2 | PNs 5-shot a=8  |   960 |  60480 |     	    61440  |
> > |	3 | PNs 5-shot a=32 |  2160 |  32400 |     	    34560  |
> > |	4 | PNs 1-shot a=8  |   448 |  28224 |     	    28672  |
> > +------+------------------+-------+--------+----------------+
> > ~~~
> > First, notice that the ranking can almost be fully explained by the number of total pairs in the right column, except for 5-shot a=16 and 5-shot a=8.
> >
> > [continues below]

---

> > > ### Author Response · Authors · 2020-11-20
> > > **Answer to AnonReviewer4 - part 3/3**
> > >
> > > At first glance, this does not fully support the hypothesis. However, we can see that the number of positive pairs is much higher in a=16 than in a=8. Since the positive pairs constitute a less frequent (and potentially more informative) training signal, this can explain the difference. The a=32 variant has an even higher number of positives than a=16, but the loss in performance there could be explained by a drastically lower number of negatives, and by the fact that the number of ways used during training is lower.
> > > So, while indeed generally speaking the higher number of pairs the better (which is also corroborated by Figure 3, as we move right on the x-axis both the NCA and PNs performances increase), one should also consider how this interacts with the positive/negative balance and the number of classes present within a batch.
> > >
> > > \> _“Can the authors elaborate on the "no S/Q" ablation (Figure 4, row 7)? What is the point of reference when computing distances for support and query examples? Is the loss computed in the same way for support and query examples?”_
> > >
> > > The way the ablation is performed is as follows: we take all the images in the query set, and all the prototypes in the support set. Then, the NCA loss is computed on the union of the query points and prototypes - there is no difference in loss computation between both sets. Equation (5) in Appendix A.2 formulates the loss precisely. We apologise if this was not clear, and we have adapted the text accordingly.
> > >
> > > \> _“Wouldn’t it be conceptually cleaner to compute leave-one-out prototypes? [..] In my mind, this would be the best way to remove the support/query partition while maintaining prototype computation”_
> > >
> > > We thought  that the current form of the ablation is a simple way to go from 5-shot prototype training to the “no S\Q” ablation without changing the batch setup drastically. In the current form,  the combination of the “no prototype” ablation with  the “no S\Q” ablation naturally leads to the NCA, as shown in Equations (4), (5), and (6). We think this additive property of the ablation is important in showing the independent effect of each one.
> > >
> > > The ablation suggested by the reviewer is neat, and does remove the support/query separation while maintaining prototype computation. However, due to its leave-one-out nature, all possible positive distance pairs will be seen in the loss function. This is because the distance of each point will be computed to prototypes consisting of the remaining points in the set of images from the same class, which essentially recovers all of the gradient signal. Part of what we wanted to investigate from the ablation is whether generating prototypes in the support set also leads to loss of training signal, as there are no positive pairs being computed between images belonging to the same prototype.
> > >
> > > **Additional feedback**
> > >
> > > \> _“I haven’t encountered an implementation that uses a flag to differentiate between support and query examples.”_
> > >
> > > We believe the term “flag” is unclear and creates confusion - we modified the text accordingly.
> > > What we meant is that each sample in a batch is either belonging to the support or the query set, and that these are mutually exclusive (and thus can be represented as a flag).
> > >
> > > \> _“Benchmarks like Meta-Dataset evaluate on variable-ways and variable-shots episodes”_
> > >
> > > Thanks for pointing this out - we corrected the sentence accordingly.
> > >
> > > \> _“I’m not too concerned with the computational efficiency of NCA.  The pairwise Euclidean distances can be computed efficiently using the inner- and outer-product of the batch of embeddings with itself.”_
> > >
> > > We agree, and that’s the implementation we used to efficiently compute pairwise Euclidean distances. We opted for writing the remark on computational efficiency because we thought someone could be interested in possible practical scenarios far from few-shot classification, such as segmentation (where samples correspond to pixel embeddings), or scenarios with a strict computational budget (e.g. real-time processing).
> > >
> > > We hope that the above answers address the reviewer's comments - we are happy to discuss further.
> > >
> > > * [Chen et al.] A Closer Look at Few-shot Classification - ICLR 2019
> > > * [Snell et al.] Prototypical Networks for Few-shot Learning - NeurIPS 2017
> > > * [Wang et al.] SimpleShot: Revisiting Nearest-Neighbor Classification for Few-Shot Learning - arXiv:1911.04623

---

> > > > ### Comment · AnonReviewer4 · 2020-11-25
> > > > **Satisfied with the authors' response**
> > > >
> > > > I would like to thank the authors for their detailed response. I feel like my main concerns have been addressed, and I increased my score accordingly.

---

### Official Review · AnonReviewer2 · 2020-10-28
**Interesting work, unsupported claims**

**Rating:** 4
**Confidence:** 5

**Review:**

The paper's starting point is the question whether the episodic training is beneficial, or not, for FSL / Prototypical Networks. The work can be seen as a follow-up of the recent works showing that simple baselines can outperform rather sophisticated few-shot learning models. Towards answering this question, this paper points out that Prototypical Networks (PN) are related to Neighborhood Component Analysis (NCA), and NCA can be considered as an episodic training-free alternative of PN.

In more detail, PN aims to learn per-class prototypes based on sample averaging in the feature space. NCA, in contrast, aims to maximize the ratio of total similarity between same-class example pairs to the total similarity between different-class pairs. Due to their similarities in terms of their formulations, the paper claims that NCA loss can be considered as an alternative to PN loss to do non-episodic representation learning for few-shot learning purposes. In addition, the paper has a few strong claims, such as episodic training is “detrimental to learning” and “under no circumstance beneficial to differentiate between support and query set within a training batch". Clearly, these are intriguing claims.

However, there is a gap between the claims and the experimental validation. First, even if ProtoNet loss and NCA loss seem to be similar to each other, they're nevertheless different models, and it takes quite a significant manipulation to convert PN to NCA. Therefore, the fact one particular non-episodic-training based model gives superior results compared to those of PN with episodic-training, does tell us much about detrimental effects of episodic training for PN or in general. Second, while the paper's observations that NCA has the advantage of using more pairwise similarities within a batch compared to PN is indeed insightful, it rather points out to certain weaknesses in the way per-batch / per-episode data is being utilized by the PN formulation, instead of problems about episodic training.

Overall, the paper has interesting observations about PN's weaknesses and shows why one particular simple non-episodic training / non meta-learned approach (NCA) can yield superior results compared to PN, which is a relatively mode sophisticated & well-established approach. However, the paper's (over-strong) claims remain mostly unsupported, which makes the otherwise interesting work poorly framed. The paper, with more water-tight arguments only, could otherwise be a valuable contribution but it requires quite significant & fundamental revisions throughout the paper, therefore, is not ready for publication in its current form.

Post-rebuttal: I would like to thank for the detailed responses and the careful revisions made in the paper. Overall, the paper is now definitely improved in certain ways and initiates an important discussion on the value of episodic training & classifier synthesis for few-shot learning, as opposed to typically-simpler metric-learning based approaches. The paper also approaches this problem from an interesting point of view, by focusing on sample utilization in the episodic training of PN.

However, I still find that the the paper remains somewhat weak in its current form for the following reasons:
- I maintain my view that NCA vs PN are not direct alternatives to each other, considering that PN allows learning a representation that is optimized for class-average to sample comparisons, whereas, NCA uses a sample-to-sample distance based loss. The fact that the very construction of these two models, despite the similarities pointed out, blurs the strength of the overall NCA vs PN based discussion on the value of episodic training.
- The claims about the advantages of non-episodic training of NCA is mainly based on the observation that NCA creates more positive/negative labels. However, it is not clear whether it is the more efficient utilization of training samples or just the differences in terms of the predictive model formulations. (Perhaps, averaging based prototype computation is a bad idea, after all, which may not have directly anything to do with episodic training.)
- To this end, Fig. 3 is indeed interesting, but again the results are not very clear. Here, careful optimizer re-tuning specially for each case can be necessary as subsampling degrades the gradient approximation quality, which creates the question how much fundamentally important efficiency in batch utilization is, as long as one uses a proper optimizer.
Overall, I think the paper makes a valuable step in an interesting direction but the paper fails to make a strong-enough case. Overall, I  improve my rating by a single level to 4, but find that the paper is not stronger than this in its current form.

---

> ### Author Response · Authors · 2020-11-20
> **Answer to AnonReviewer2 - part 1/2 - we believe the low score might come from a misunderstanding around what we intend to claim**
>
> We thank this reviewer for their time and comments, to which we answer below.
>
> \> _“The paper's (over-strong) claims remain mostly unsupported, which makes the otherwise interesting work poorly framed.“_
>
> \> _“There is a gap between the claims and the experimental validation.”_
>
> We would kindly ask this reviewer to be more specific - in what way are our claims mostly unsupported? Reading the review, it seems that the reviewer concern is that we are claiming our findings are applicable to all methods making use of episodic learning. This is not our intention, and we qualify our work multiple times, starting from the title. However, we understand where we might have been unclear, and we propose a simple way to improve the clarity of the text below.
> We would like to address two partial quotes from our abstract that are mentioned in the second paragraph of this review, as this reviewer has called them _“intriguing”_, but _“strong”_.
>
> **1.**  _“we investigate the usefulness of episodic learning in Prototypical Networks [..] it is under no circumstance beneficial to differentiate between support and query set within a training batch."_
> We did evaluate PNs in a variety of settings and, despite the fact that our implementation of PNs has a higher performance than the ones reported in recent literature (e.g. [Wang et al.], [Chen et al.]), we found it to always be outperformed by NCA (that’s what we meant with “under no circumstances”).
> However, we believe the “under no circumstance” qualifier might be misinterpreted, as it could sound like we are referring to all the algorithms making use of episodic training. This is not our intention and we thank the reviewer for pointing this out.
> We changed the sentence to _“we found that, for Prototypical Networks, it is detrimental to use the episodic learning strategy of separating training samples between support and query set, as it is a data-inefficient way to exploit training batches.”_
> To avoid possible ambiguities, we have also edited a sentence in the last paragraph of the Introduction, specifying again that the conclusions regard Prototypical Networks (while in the submitted version this was implicit from the sentence before).
>
> **2.** _“[episodic training] is detrimental to learning”_.
> We wrote that, for Prototypical Networks, episodic learning is detrimental to _performance_, and we believe we provided arguments and data to justify this.
> We showed it empirically on a number of different setups (Fig.2, Fig.4) and we show that the episodic strategy of PNs is not better than using the NCA and randomly discarding pairs from within a batch (Fig. 3).
> Moreover, we show the intuition behind why this is the case by analysing the losses (eq. 2 to 6) and using a simple visualization to better illustrate the concept (Fig.1)
>
> [continues below]

---

> > ### Author Response · Authors · 2020-11-20
> > **Answer to AnonReviewer2 - part 2/2**
> >
> > \> _“While the paper observation [..] is indeed insightful, it rather points out to certain weaknesses in the way per-batch / per-episode data is being utilized by the PN formulation, instead of problems about episodic training.”_
> >
> > We believe this comment arises from the misunderstanding from point 1 above, because the _“way per-episode data is being utilized by the PN formulation”_ is what we mean when we say that PNs use episodic training.
> > PNs’ “inner loop” [Hospedales et al, 2020] amounts to no parameter adaptation, and thus the “episodic” nature of PNs is limited to how episodes are sampled and what role each point within an episode has (support points have to form prototypes and their distances to corresponding query points have to be minimised).
> > Hence, we think that this reviewer might believe that we are claiming that our results apply to the entire episodic learning literature. This is not the case, as we qualified the results being specific to PNs throughout the text. We hope that the two re-worked sentences detailed in the previous answer clarify the confusion.
> >
> > Moreover, please note that, following the suggestion of AnonReviewer4 (the second reviewer), we applied the ablation analysis of Fig.4 also to Matching Networks [Vinyals et al.], showing analogous results to the ones obtained for PNs: the separation of roles between images in the support and query sets, typical of episodic learning, is detrimental to the performance of not only PNs, but also Matching Networks.
> > Clearly, this does not extend the considerations of this paper to algorithms that have not been tested, but we believe these are important results, and that they are timely within this community.
> >
> > \> _“It takes significant manipulation to convert PN to NCA”_
> >
> > \> _“The fact one particular non-episodic-training based model gives superior results compared to those of PN with episodic-training, does [not] tell us much about detrimental effects of episodic training for PN.”_
> >
> > The similarity between PNs and NCA is stated by the very authors of PNs [Snell et al.], who consider it as the closest method in their related work section: _“our method is most similar to the non-linear extension of NCA [..] A key distinction [..] is that we form a softmax directly over classes, rather than individual points”_.
> > Starting from the PNs loss, the NCA loss is an intuitive “episode-free” baseline because it does not rely on the roles of query and support set.
> > Importantly, this frees us from the burden of tuning the three (hyper-)parameters {ways, queries, shots} that need to be specified to construct an episode.
> >
> > We hope that our comments address the concerns this reviewer has - we are happy to discuss further.
> >
> > * [Chen et al., 2019] A Closer Look at Few-shot Classification - ICLR 2019
> > * [Hospedales et al.] Meta-Learning in Neural Networks: A Survey - arXiv:2004.05439
> > * [Snell et al.] Prototypical Networks for Few-shot Learning - NeurIPS 2017
> > * [Vinyals et al.] Matching Networks for One Shot Learning - NeurIPS 2016
> > * [Wang et al.] SimpleShot: Revisiting Nearest-Neighbor Classification for Few-Shot Learning - arXiv:1911.04623

---

### Author Response · Authors · 2020-11-20
**General response**

We thank all the reviewers for their time and valuable comments.

We answered each reviewer in-thread, and we have edited our submission incorporating the suggestions.
(Apologies for the deleted messages in the OpenReview forum, we made a mistake with the markdown).

Below a list of the main changes:
* As suggested by AnonReviewer4, we have included results for Matching Networks in Appendix A.7. We show that the episodic batch setup is detrimental in this case as well, which corroborates the findings of the main paper.
* We have extended the analysis on the number of pairs exploited by different batch strategies (Appendix A.8), showing that NCA can exploit a number of extra pairs that grows as $O(w^2(m^2+n^2))$ (where $w$, $m$ and $n$ are the number of ways, queries and shots respectively).
* We have included detailed comments on how we chose the (non-episodic) hyper-parameters for PNs and NCA to ensure a fair comparison (Appendix A.5)
* We have reported and discussed the ablation experiments also for batch sizes 128 and 512 (Appendix A.7).
* In Sec 4.1, we have commented on the influence of outliers in the performance gap between the Soft Assignment and 1-NN with Centroid.
* We have also made small edits throughout the text, to address the reviewers suggestions and to improve the clarity of some of our points.

---

### Decision · Program_Chairs · 2021-01-07
**Final Decision**

**Decision:**

Reject

**Comment:**

This paper is right on the borderline. It questions the utility of episodic training from a novel perspective, driven by a comparison to NCA, with thorough experiments. The hypothesis that more pairwise comparisons per batch/episode benefit learning is also quite interesting, but some reviewers didn’t feel this was convincingly presented.

Prototypical networks are indeed a popular method for FSL, but I do as well think that NCA is more closely related to matching networks, and that it makes more sense for that to be the focus of experimentation. Matching networks involve more direct pairwise comparisons, and so a leave-one-out baseline with this model would probably be a useful comparison.

While I appreciate the desire to focus on a fundamental aspect of FSL and not chase state of the art, I think that it’s important to show where one should go from here. That is, as the reviewers pointed out there are many mechanisms beyond vanilla PNs that have yielded better results than those presented in this paper. I don’t think matching SOTA is necessary here, but it would be nice to show that the insights here complement other mechanisms in FSL.